# *E*-Stilbenes: General Chemical and Biological Aspects, Potential Pharmacological Activity Based on the Nrf2 Pathway

**DOI:** 10.3390/ph17020232

**Published:** 2024-02-09

**Authors:** Elaine L. S. S. Mendonça, Jadriane A. Xavier, Marilene B. T. Fragoso, Messias O. Silva, Pierre B. Escodro, Alane C. M. Oliveira, Paolo Tucci, Luciano Saso, Marília O. F. Goulart

**Affiliations:** 1Program of the Northeast Biotechnology Network (RENORBIO), Institute of Chemistry and Biotechnology, Federal University of Alagoas (UFAL), Maceió 57072-900, Brazil; elaine.mendonca@fanut.ufal.br (E.L.S.S.M.); messias.silva@iqb.ufal.br (M.O.S.); 2Institute of Chemistry and Biotechnology, UFAL, Maceió 57072-900, Brazil; jadrianexavier@iqb.ufal.br (J.A.X.); marilene.tenorio@iqb.ufal.br (M.B.T.F.); 3Veterinary Department, UFAL, Maceió 57072-900, Brazil; pierre.vet@gmail.com; 4College of Nutrition, UFAL, Maceió 57072-900, Brazil; alane.oliveira@fanut.ufal.br; 5Department of Clinical and Experimental Medicine, University of Foggia, 71121 Foggia, Italy; paolo.tucci@unifg.it; 6Department of Physiology and Pharmacology “Vittorio Erspamer”, Sapienza University, 00185 Rome, Italy

**Keywords:** antioxidants, anti-inflammatory, polyphenol, molecular mechanisms

## Abstract

Stilbenes are phytoalexins, and their biosynthesis can occur through a natural route (shikimate precursor) or an alternative route (in microorganism cultures). The latter is a metabolic engineering strategy to enhance production due to stilbenes recognized pharmacological and medicinal potential. It is believed that in the human body, these potential activities can be modulated by the regulation of the nuclear factor erythroid derived 2 (Nrf2), which increases the expression of antioxidant enzymes. Given this, our review aims to critically analyze evidence regarding *E*-stilbenes in human metabolism and the Nrf2 activation pathway, with an emphasis on inflammatory and oxidative stress aspects related to the pathophysiology of chronic and metabolic diseases. In this comprehensive literature review, it can be observed that despite the broad number of stilbenes, those most frequently explored in clinical trials and preclinical studies (in vitro and in vivo) were resveratrol, piceatannol, pterostilbene, polydatin, stilbestrol, and pinosylvin. In some cases, depending on the dose/concentration and chemical nature of the stilbene, it was possible to identify activation of the Nrf2 pathway. Furthermore, the use of some experimental models presented a challenge in comparing results. In view of the above, it can be suggested that *E*-stilbenes have a relationship with the Nrf2 pathway, whether directly or indirectly, through different biological pathways, and in different diseases or conditions that are mainly related to inflammation and oxidative stress.

## 1. Introduction

Stilbenes account for a vast group of polyphenols characterized by a 1,2-diphenylethylene (C6–C2–C6) skeleton [1]. The medicinal properties of stilbenes, in the human body, can be attributed to their peculiar chemical structures. These include the ability to oligomerize, which increases the interaction with immune components; the presence of the oxidizable catechol group (piceatannol, PIC), which can enhance anti-inflammatory properties; and the presence of two methoxy groups (pterostilbene, PTS), which allow potent antioxidant activity [2,3].

Studies available in the literature have focused on investigating whether these medicinal or nutraceutical properties can be modulated by stilbenes’ molecular action in the nuclear factor pathways, which account for regulating the expression of proinflammatory cytokines and reactive oxygen species (ROS) [4,5].

Two main pathways have been explored to date, namely, the nuclear factor kappa B (NF-κB) pathway and the nuclear factor erythroid-derived 2 (Nrf2) pathways. The NF-κB signaling pathway results from interactions between dimeric transcription factors (NF-κB-inhibitory regulators (IκBs) and the IκB kinase complex (IKK)) that regulate genes involved in human immunological and inflammatory responses. The activation of this pathway is associated with an increase in proinflammatory cytokines, interleukin 1β (IL-1β), interleukin-6 (IL-6), tumor necrosis factor α (TNF-α), and adipokines, among others, as well as ROS levels; consequently, it is associated with some acute or chronic diseases [6,7].

Nrf2 is a basic leucine zipper protein (bZIP) that plays an essential role in various processes, such as xenobiotics’ detoxification, heme group metabolism, antioxidant enzymes’ coding catalase (CAT), superoxide dismutase (SOD), glutathione peroxidase (GPx), and inflammatory genes’ suppression. It plays a fundamental role in maintaining cell homeostasis and mitigating different diseases [8,9]. However, the activation of these pathways requires precursors capable of triggering the signaling cascade, such as oxidative stress enhancers (NF-κB) or antioxidants like stilbenes (Nrf2) [6,7,8,9].

Current studies focus, mainly, on investigating the Nrf2 signaling pathway modulated by stilbenes because, although substantial findings about this nuclear factor are available in the literature, their molecular mechanisms and therapeutic potential to be applied in individuals with chronic and metabolic diseases need further study. Considering this, the aim of the current review was to critically analyze evidence regarding stilbenes’ action in both the human metabolism and the Nrf2 activation pathway, with an emphasis on inflammatory and oxidative stress aspects associated with the pathophysiology of chronic and metabolic diseases.

## 2. Stilbenes

In chemical terms, stilbenes (non-flavonoid polyphenolic class) share a common structure characterized by a 14-carbon skeleton (C6–C2–C6) with two benzene rings linked by an ethylene bridge (Figure 1) [10,11]. Overall, one of the aromatic rings in their structures carries two hydroxyl groups, whereas the other aromatic ring can carry both hydroxy and methoxy groups in different positions [12]. The central ethylenic portion enables two stereoisomers, namely, trans-stilbene (*E*-stilbene–the natural, stable, and most common of the two) and cis-stilbene (*Z*-stilbene, less stable) [13].

Stilbenes can be represented by a variety of chemical compounds, some of which have been shown to have biological effects, including resveratrol (RESV), PIC, PTS, polydatin (PDT), stilbestrol (DHS), and pinosylvin (PIN). Natural stilbenes can be found in their free (mostly), glycosylated, prenylated, and methoxylated forms. In addition, they can be monomeric, dimeric, trimeric, and polymeric, which are features capable of affecting their biological activity [12].

Glycosylation is identified as the most frequent change observed in secondary metabolites that can also modify stilbenes’ physical, chemical, and biological properties [14]. This process enables higher stilbene stability and protection against enzymatic reactions by increasing their mean lifetime and their solubility in aqueous media, which preserves their biological properties and enables their transport to different organs [15]. It is worth mentioning that stilbenes are usually stored in their glycosylated form, such as PDT (RESV’s glycosylated form), which can reach a concentration approximately six times higher than that of its free form (RESV) [16]. Glycosylation takes place in plants via glycosyltransferases (GTs) by acting in activating sugars’ (single or multiple) transfer between nucleotide donors (uridine diphosphate (UDP)-glucose, for example) and plants’ molecular receptors [14].

Methylation reactions of phenolic hydroxy, catalyzed by S-adenosyl-L-methionine (SAM)-dependent O-methyltransferases (OMTs) [11], lead to methoxy-stilbenes’ formation, notably, PIN monomethyl ether (3-hydroxy-5-methoxy-stilbene) and PTS (3,5-dimethoxy-4′-hydroxy-stilbene) [17,18]. This reaction can influence stilbenes’ solubility and reactivity, and it can negatively (PIN monomethyl ether’s antifungal and antibacterial activity are lower than those of its free form) and/or positively (higher anticancer activity is associated with RESV methylation) affect their biological activity [19,20,21]. It is known that hydroxy group methylation increases PTS lipophilicity in comparison to RESV; this process leads to greater bioavailability, a fact that justifies the increase in pharmacological interest [22].

Stilbenes’ prenylation takes place in some plant species, such as *Macaranga* spp., *Glycyrrhiza* spp., *Morus alba*, and *Arachis hypogaea*, via prenyltransferase [23,24,25,26]. Despite their biological relevance, their biosynthetic pathways are yet to be fully understood; however, it is known that, in comparison to their nonprenylated counterparts, stilbenes present higher bioavailability due to increased lipophilicity linked to the prenyl groups [11]. Prenyltransferases are in chloroplasts and are specific to the prenylation of stilbenes [26,27,28].

Multifactorial conditions (isomeric geometric forms *Z*- and *E*-; reaction processes; and the stilbene’s type) can interfere with stilbenes’ activity, storage, concentration, biological activity, and bioavailability and can also affect the secondary metabolites’ medicinal and pharmacological beneficial relevance to modulate health and disease processes. Despite some gaps observed in the scientific knowledge, benefits provided by stilbenes, mainly by RESV, PIC, PTS, PDT, DHS, and PIN, are indisputable. Thus, to improve the comprehension of the fundamental mechanism and make rational assumptions about stilbenes’ potential effects on human health, the present investigations have concentrated on examining their biosynthesis and metabolic processes.

### 2.1. Stilbenes’ Biosynthesis

Stilbenes are secondary metabolites produced by plants to help protect them in response to certain external aggressors (ultraviolet radiation; cracks; fungal, viral, or bacterial attacks; and pesticides, among others). They were initially identified as phytoalexins (defensive substances produced in response to infections), belonging to the class of polyketides [29]. Their *Z*- and *E*-isomers accumulate in vegetables’ peel during plants’ developmental stages. This occurs because, in anatomical terms, the peel is a plant’s outermost portion; therefore, it is more susceptible to both biotic and abiotic stressors/aggressors [10,13].

Stilbene biosynthesis can take place either in plants or in microorganisms [10,11,12]. The route identified in plants is the route most often documented for stilbenes and DHS’s biosynthesis, which has shikimate as a precursor. Shikimate, in turn, generates two aromatic amino acids, phenylalanine and tyrosine, which play essential roles in trans-cinnamic acid, or its p-coumaric derivative formation, to initialize the phenylpropanoid pathway accountable for the biosynthesis of several primary and secondary metabolites, such as stilbenes, flavonoids, coumarins, hydrolysable tannins, monolignols, and lignans (Figure 2) [30].

Given this complex cascade for stilbenes’ biosynthesis and low content production, researchers developed a strategy to increase their synthesis by using plant cell cultures, based on the assumption that biosynthesis is triggered by external stressors or aggressors such as ROS, methyl jasmonate (MeJA), and salicylic acid (SA) [31]. Based on the same line of investigation, microbiologists have tried to mimic stilbenes’ biosynthetic pathways in heterologous organisms [32,33].

**Figure 2 pharmaceuticals-17-00232-f002:**
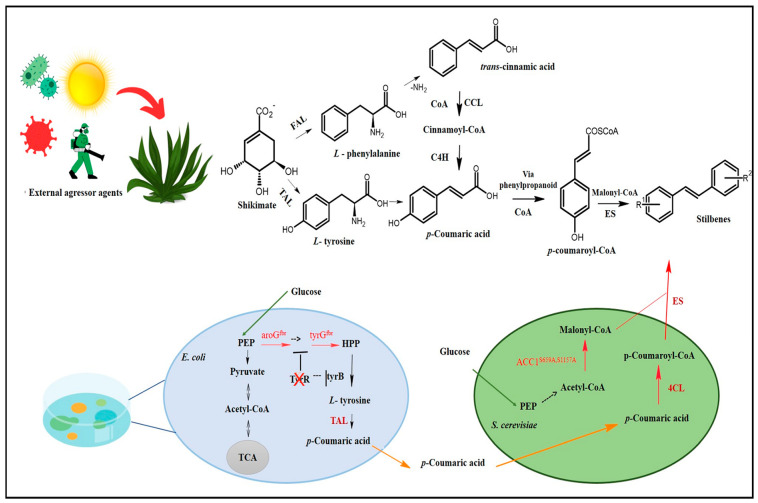
Stilbenes’ biosynthesis route. Adapted from Yuan [34]. CoA: coenzyme A; COSCoA: acyl CoA thioester; FAL: phenylalanine ammonia-lyase; TAL: tyrosine ammonia-lyase; CCL: cinnamate CoA ligase; C4H: cinnamate-4-hydroxylase (cytochrome P450); ES: stilbene synthetase; HCO_3_^−^: bicarbonate ion; ATP: adenosine triphosphate; *E. coli*: *Escherichia coli*; *S. cerevisiae*: *Saccharomyces cerevisiae*; aroG^fbr^: feedback-inhibition-resistant 3-deoxy-d-arabinoheptulosonate-7-phosphate synthase; tyrA^fbr^: feedback-inhibition-resistant chorismate mutase/prephenate dehydrogenase; ACC1^S659A,S1157A^: inhibition-resistant acetyl-CoA carboxylase; PEP: Phosphoenolpyruvate; HPP: 4-hydroxyphenylpyruvate; tyrB and tyrR: transcriptional regulator; TCA: tricarboxylic acid cycle; 4CL: 4-coumarate-CoA ligase; R1: Substituent 1; R2: Substituent 2.

This revolutionary and innovative field has awakened a new aspect of multidisciplinary research according to which several scientific fields interact with each other to optimally meet the demand to produce secondary plant metabolites through microorganisms and cell cultures. This encouraged the development of new techniques based on metabolic engineering in cellular factories. Yuan et al. [34], for instance, used a co-culture system comprising *Escherichia coli* and *Saccharomyces cerevisiae* for RESV biosynthesis, which reached 36 mg/L, whereas Yan et al. [35] managed to biosynthesize 80 mg/L of PTS based on the introduction of genes capable of encoding transcription activator-like (TAL) protein as well as other enzymes and substrates accountable for PTS biosynthesis in *Escherichia coli*. However, it is necessary to conduct biological studies in vitro and/or in vivo to assess toxicity and safety concerning stilbenes synthesized through microbiological organisms.

### 2.2. Stilbenes’ Biological Metabolism

Although stilbenes have metabolites that share a common skeleton and demonstrate similar biological activity, the in vivo metabolism of the majority of these compounds remains unknown [36]. Each stilbene has its own functional and structural specificity based on reactions involved in its biosynthesis. This factor reflects its unique behavior, which can be observed in pharmacokinetic studies involving ADME (absorption, distribution, metabolism, and excretion) mechanisms [36,37]. However, studies focused on this research perspective do not yet have enough support, since the dose and administration route set for each stilbene are not yet standardized [3,12].

Phases I and II of biotransformation can be accessed by drugs, medical goods, and/or bioactive substances; of these, stilbenes preferentially travel through phase II due to their bioactivity [36,38,39]. Similar natural processes, including glucuronidation, sulfation, and intestine biotransformation, are shared by stilbenes (Figure 3) [36,37,40,41,42,43]. Sulfation is another important stilbene metabolism pathway, since it is mostly accountable for metabolites’ excretion. This pathway has been associated with stilbenes’ anticancer and cardioprotective activity [44].

However, in addition to the hepatic pathways listed above, there has been increasing evidence of intestinal biotransformation pathways, mainly associated with RESV. RESV produces dihydroresveratrol, which, in turn, is a metabolite deriving from the intestinal microbiota [41,42,45,46]. Stilbenes delivered through oral or intravenous routes show significantly low concentrations which can range from nano to micromolar; however, they still demonstrate significant biological effects. Experimental results have shown intense intestinal and hepatic biotransformation metabolism 1 h after RESV administration, as observed in a serum RESV concentration ranging from 0.3 to 2.4 µmol/L, whereas its glucuronidated and sulfated metabolites recorded concentrations approximately 20 times higher [47].

Bode et al. [48] assessed human fecal samples collected after RESV supplementation and identified two bacterial strains involved in RESV biotransformation, namely, *Slackia equolifaciens* and *Adlercreutzia equolifaciens*. In addition to causing dihydroresveratrol formation, these strains generated 3,4′-dihydroxy-trans-stilbene and 3,4′-dihydroxybisbenzyl one (lunularin). However, intestinal bacteria capable of producing dehydroxylated metabolites were not identified, although hydroxy groups’ cleavage plays a key role in the microbial transformation of various compounds, such as lignans, as well as phenolic and bile acids. Therefore, interindividual variability influences intestinal microbiota composition. Nevertheless, external factors, such as diet and physical activity, cannot be ruled out.

On the other hand, Sun et al. [49] investigated PTS metabolites in CD-1 mice’s colonic contents and mucosa. Animals subjected to diet supplemented with PTS for 3 weeks presented pinostilbene metabolite formation. In addition, the concentration of this metabolite in the analyzed colonic content was approximately 10 times higher than that observed for the colonic mucosa. However, it is necessary to conduct further studies to help fill this knowledge gap, since it has been suggested that the intestinal microbiota accounts for PTS demethylation. This result was corroborated by previous investigations associated with microbial demethylases involved in flavonoid, anthocyanin, and lignan demethylation processes, among others [50]. It is worth mentioning that this on-site activity can optimize treatments for dysbiosis and inflammatory bowel diseases (IBDs) such as ulcerative colitis (UC) and Crohn’s disease (CD) [41].

Pharmacokinetics have a significant impact on the experimental data of stilbenes in vivo. In fact, RESV has a low oral bioavailability: less than 30% in a rat model [51] and less than 0.5% in humans [52]. When RESV reaches the colon, it travels to the enterocytes, where it is sulfated (by the SULT1A1) and glucuronidated (through UGT1A1 and UG-TA9). The enterocytes release intact RESV and its metabolites into the portal circulation, where they are transported to the liver, being further conjugated by the same enzyme that was present in the enterocytes [53].

In conclusion, a small fraction of intact RESV and its metabolites enter systemic circulation and are absorbed by peripheral tissues. Conjugated RESV is involved in enterohepatic circulation. Additionally, some of the conjugated metabolites and RESV pass from the small to the large intestine, where the gut microbiota can process them to produce dihydro-resveratrol (DHR), lunularin (L), and 3,4′-dihydroxy-trans-stilbene [48,53].

Despite RESV’s relatively limited bioavailability, several investigations have shown that it has biological activity in vivo in a wide variety of animal trials. Given the presumably non-physiological doses and the exclusion of the role played by RESV metabolites, studies conducted in vitro have shown a wide variety of controversial biological effects [54]. Some scholars believe that metabolites can store stilbenoids [52,55].

PTS is more lipophilic and metabolically stable due to the presence of two methoxy groups, although only one of them is accessible for glucuronidation or sulphation purposes [56]. In fact, RESV was more frequently metabolized by glucuronidation rather than by PTS in human liver microsomes [57]. That study demonstrated gender-based differences in stilbene metabolism. The highest bioavailability rate was recorded for PTS [56], followed by PIC [58]. The lowest oral bioavailability rate was recorded for gnetol (2,3′,5′,6-tetrahydroxy-trans-stilbene) [59], but it has shown a half-life longer than values reported for RESV [51] and PTS [60] after oral administration in rats.

However, despite the encouraging results, further exploration of pharmacokinetic aspects is required, since there are a lack of data on the metabolism and biotransformation of most stilbenes. Future research should investigate these routes, i.e., routes enabling biological activity such as that of nuclear factors (Nrf2 and NF-κB).

## 3. Stilbenes: Diseases-Based Biological and Pharmacological Activities

Nowadays, we are aware of a good number (almost 100) of stilbene derivatives that have a wide variety of biological effects on several experimental models [12].

Anticancer, antimicrobial, antidiabetic, cardioprotective, anti-inflammatory, antioxidant, and neuroprotective actions are biological effects of these compounds that have been described in the literature. This broad range of biological consequences undoubtedly involves a wide variety of action mechanisms.

### 3.1. Anticancer

Stilbenes’ anticancer effect appears to depend on blocking a wide variety of signaling pathways involved in tumor growth, as well as in certain cytochrome P450 isoforms, to prevent the metabolic activation of procarcinogens [61,62]. RESV and PTS have shown significant anticancer properties [63]. These compounds inhibit topoisomerase 1 activity, as well as the DNA damage-repair pathway mediated by tyrosyl-DNA phosphodiesterase 1, which accounts for tumors’ resistance to drugs [64]. The RESV methylated derivative 3,5,4′-trimethoxystilbene was capable of inhibiting Caco-2 cells’ growth in human colon cancer, as well as tubulin polymerization, in a dose-dependent manner [65]. Another group of researchers reported that this compound has shown potential antitumor activity in a breast cancer cell model by downregulating phosphatidylinositol 3-kinase/AKT signaling (PI3K-AKT) [66].

A synthetic analog of RESV, named trans-4,4′-dihydroxystilbene (DHS), acted as strong DNA replication inhibitor in mouse models subjected to tumor xenografts and showed effects against pancreatic, ovarian, and colorectal cancer cells [67]. DHS induced cyclin F-mediated downregulation of ribonucleotide reductase regulatory subunit M2 of ribonucleotide reductase (RRM2) by proteasome. Moreover, it was observed to reduce ribonucleotide reductase activity and decrease deoxyribonucleoside triphosphates synthesis through concomitant DNA replication inhibition, cell cycle arrest at S-phase, DNA damage, and, finally, apoptosis [67]. Cyclin is a family of proteins accountable for controlling the progression of a given cell throughout the cell cycle by activating cyclin-dependent kinase (CDK) enzymes. These proteins are also the target of RESV, hemsleyanol D, and (+)-α-viniferin (isolated from the plant species *Shorea roxburghii*), which have significantly decreased cyclin B1 expression; cyclin B1, in its turn, also suppressed cell cycle progression [68].

A methoxylated stilbene, named isorhapontigenin, induced cell death and cell growth arrest in breast cancer models by activating the caspase pathway [69]. PIC has effects similar to that of RESV on a wide variety of target sites [70,71]. PIC appears to have stronger anticancer activity than RESV, likely because its hydroxyl group is in the ortho-position instead of the meta-position [72]. Hepatocellular carcinomas are among several systemic malignancies whose tumor growth can be slowed by PIC [73]. Cell cycle arrest, modulation of proteins involved in apoptosis regulation, caspase (-3, -7, -8, and -9) activation, mitochondrial potential loss, and cytochrome c release are the mechanisms accounting for mediating anticancer actions [71]. Moreover, PIC inhibits the activation of several transcription factors such as NF-κB, which is a crucial transcriptional regulator activated in response to cell stress [71].

PIN modulates cancer cell growth inhibition and death by controlling the overexpression of TGF-β superfamily member NAG-1 (nonsteroidal anti-inflammatory drug-activated gene), which is linked to tumor progression and development processes [74,75]. In addition, PIN inhibits metastatic oral cancer cells by controlling both metalloproteinase (MMP-2) expression and activity via the ERK pathway (proteins linked to mitogen-activated protein kinase (MAPK) pathway activation, a common incidence in carcinogenesis cases) [76], whereas AMP-activated protein kinase α1 (AMPKalpha1) downregulation is the mechanism leading to leukemia cell death [77].

### 3.2. Antimicrobial

Stilbenes’ antimicrobial activity is not surprising, since they are substances produced by plants to function as toxins against attacking organisms [78]. These antimicrobial effects were attributed to damage in both the microbial cell wall and cell membrane, to cytoplasm condensation, and to membrane potential disruption.

RESV has shown activity on Gram-negative bacteria, although this activity was lower than that of PIN, which, similarly to PTS, showed higher activity against Gram-positive bacteria [79,80]. Both PIN and PIC presented clear antimicrobial activity through outer membrane destabilization in Gram-negative microorganisms as well as through interactions with cell membrane [80]. RESV, PIN, PIC, and PTS were also active against fungi [81]. Their activity against fungal pathogens was attributed to downregulation of both the ergosterol biosynthesis and the Ras/cAMP pathway, which plays an essential role in controlling and integrating growth, cell cycle progression, and metabolic activity [82].

Recently, kobophenol A, which is a stilbenoid isolated from the *Caragana* genus, was shown to be capable of blocking the interaction between the ACE2 receptor and the spike receptor binding domain (S1-RBD) of SARS-CoV-2.83 Kobophenol A, and *Caragana sinica* extracts were previously tested to prevent and treat West Nile virus infection; they demonstrated antiviral activity by inhibiting neuraminidase (patent application no.: KR20200026550A) [83].

A stilbene glycoside (piceid-(1→6)-β-d-glucopyranoside) derived from *Parthenocissus tricuspidata* demonstrated significant blood schizontocidal activity against Plasmodium berghei in outbred male ICR mice, although its action mechanism remains unknown [84].

Longistylin A, which is an abundant stilbene isolated from *Cajanus cajan* leaves, presented strong antibacterial activity against methicillin-resistant *Staphylococcus aureus* (MRSA) in vitro in association with bacterial membrane potential disruption and increased membrane permeability. Topical treatment with longistylin A applied to skin injury in vivo improved wound healing and closure in an MRSA-infected wound-healing mouse model [85].

### 3.3. Antidiabetic

Hydrangeic acid, which is a stilbene derived from processed Hydrangea macrophylla leaves, promoted the adipogenesis of 3T3-L1 cells (fibroblast isolated from mouse embryo and often used to investigate basic cell mechanisms associated with diabetes) [86]. Hydrangeic acid has significantly increased the adiponectin amount released into the medium, 2-deoxyglucose uptake into cells, and glucose transporter 4 translocation (GLUT4). It has also increased the mRNA levels of adiponectin, peroxisome proliferator-activated receptor γ2 (PPARγ2), GLUT4, and fatty acid-binding protein (aP2), although it decreased the expression of TNF-α mRNA. Furthermore, this acid has significantly decreased blood glucose, triglyceride, and free fatty acid levels after it was orally administered to KK-Ay mice (type 2 diabetes model) for 2 weeks [87,88].

Stilbenes, such as 3,5-dimethoxy-4′-O-prenyl-trans-stilbene isolated from the Amazonian plant species *Deguelia rufescens*, as well as trans-RESV and rumexoid isolated from *Rumex bucephalophorus*, were capable of inhibiting α-glucosidase in vitro; this finding indicates their potential for use as antidiabetic drugs [89,90].

### 3.4. Cardiovascular

RESV prevents atherogenesis and promotes thrombus resistance in human vascular endothelial cells by maintaining the balance between vasodilators and vasoconstrictors (nitric oxide and endothelin). Moreover, RESV has antioxidant effects on cholesterol metabolism and prevents platelet aggregation [91]. It also lowers blood pressure in animal models in a dose-dependent manner [92]. Studies conducted with humans have shown systolic blood pressure reduction at high RESV doses [93]. A randomized double-blinded placebo-controlled trial indicated that a high PTS dose reduced both systolic and diastolic blood pressure in humans [94].

### 3.5. Anti-Inflammatory

Stilbenes can act on various inflammatory process stages and inhibition. Studies have shown that cyclooxygenase-1 (COX-1) and cyclooxygenase-2 (COX-2) were successfully inhibited by RESV, PIC, PTS, PIN, desoxyrhapontigenin, and rhapontigenin. The NF-κB pathway was suppressed by RESV, oxyresveratrol, PTS, and PIC, which reduced the release of inflammatory cytokines [95,96].

Dietary RESV decreased the death rate in an animal model of chronic dextran sodium sulphate (DSS)-induced colitis by mitigating the severity of clinical symptoms such as body weight loss, diarrhea, and rectal bleeding. RESV decreased prostaglandin E synthase-1 (PGES-1), COX-2, and inducible nitric oxide synthase (NOS) proteins’ expression by downregulating p38, which is a mitogen-activated protein kinase (MAPK) signal pathway. Furthermore, it increased anti-inflammatory cytokine IL-10 expression and decreased the expression of proinflammatory cytokines such as TNF-α and IL-1β [97].

According to a randomized clinical study, RESV has anti-TNF properties useful in the treatment of Takayasu arteritis, a chronic granulomatous inflammatory disease that affects the aorta and its major branches [98].

### 3.6. Neuroprotection

Antioxidant and anti-inflammatory activities are key components of the neuroprotective features of stilbenes. RESV protected neurons from ROS and enhanced motor coordination in a mouse model subjected to 1-methyl-4-phenyl-1,2,3,6-tetrahydropyridine (MPTP)-induced Parkinson’s [99] by scavenging hydroxyl radicals. It also protected the assessed model from dopaminergic neurodegeneration caused by lipopolysaccharide (LPS) by preventing nicotinamide adenine dinucleotide phosphate (NADPH) oxidase inhibition and microglial activation [100].

Based on a process involving proteasome, RESV promoted intracellular degradation in amyloid-beta peptides produced from several cell lines accountable for expressing wild-type or Swedish mutant amyloid-beta precursor protein 695 [101]. It also reduced learning impairment and mitigated neurodegeneration in the hippocampus of a transgenic Alzheimer’s disease mouse model by decreasing the acetylation of sirtuin 1 (SIRT1) substrates [102].

Patients with Alzheimer’s disease demonstrated a good response to RESV at doses up to 1000 mg, administered twice a day for 52 weeks, in a randomized, placebo-controlled clinical trial [103]. RESV used in this trial prevented a decrease in amyloid-beta 40 levels in the patients’ blood and cerebrospinal fluid in comparison to the placebo group, but it did not consistently affect clinical outcomes or other biomarker trajectories (including plasma Ab42, CSF Ab42, CSF tau, and CSF phospho-tau 181) [103].

Studies currently available in the literature reported that RESV improved the memory and cognition of both healthy individuals and diabetic patients with subclinical cognitive impairment, although it did not show the same effect on individuals with Alzheimer’s disease [104,105,106,107].

RESV prevented neuronal death in a rat model of global cerebral ischemia by activating PI3K-AKT signaling, as well as by decreasing glycogen synthase kinase-3 (GSK-3) and cAMP response element-binding protein (CREB) levels [108]. RESV improved cognition in an animal model of vascular dementia and increased antioxidant enzyme levels in its cerebral cortex and hippocampus, whereas malondialdehyde (lipid peroxidation product) levels decreased [109].

PTS-based treatment improved memory loss caused by streptozotocin in Sprague Dawley rats and enhanced cholinergic transmission by inhibiting cholinesterases [110]. Hydroxystilbene protected rat cortical neurons against damage caused by amyloid beta (25–35) by limiting ROS production, suppressing glutamate release, and by preventing an increase in cytosolic calcium levels [111]. Moreover, oxyresveratrol has significantly decreased brain infarct volume in a murine model of transient middle cerebral artery occlusion by inhibiting both cytochrome c release and caspase-3 activation [112].

## 4. Stilbenes’ Role in Activating the Nrf2 Pathway

The human body has pre-established regulatory mechanisms, such as nuclear factors NF-κB and Nrf2, that enable it to respond and adapt to both exogenous (such as pollution, UV radiation, pollution, physical inactivity, smoking, and alcohol consumption) and endogenous (such as cortisol, ROS, proinflammatory cytokines, hydroperoxides, and quinones) stressors [8,9,113]. Because nuclear pathways are strictly regulated, stress can cause NF-κB dysregulation, which then activates its pathway. This worsens endogenous inflammation by increasing the expression of proinflammatory agents such as chemokines, adhesion molecules, and cytokines [7,114]. Unlike NF-κB, the Nrf2 pathway can be mainly activated by key components of antioxidant systems, i.e., by direct antioxidants, which are molecules with redox-active properties capable of ruling out ROS (reduced glutathione, GSH; ascorbate; tocopherols) and enzyme systems (GPx and thioredoxin system (TXN), among others) as well as by indirect antioxidants, which are susceptible to the induction of cytoprotective genes capable of recycling and/or regenerating direct antioxidants (natural polyphenols, such as stilbenes and isothiocyanates) [115].

Despite the undeniable importance of both nuclear factors listed above, the current review focuses only on investigating the modulation of the Nrf2 pathway. After understanding its essential role in redox homeostasis, drug/xenobiotic metabolism, mitochondrial function, and deoxyribonucleic acid (DNA) repair, researchers have expanded their interest in modulating the Nrf2 pathway, mainly in the human health field. Once stimulating this pathway, the likelihood of treating or mitigating unfavorable outcomes in chronic and/or metabolic diseases increases [113]. The Nrf2 controls basal gene expression both under homeostasis and oxidative stress conditions. Furthermore, it accounts for regulating approximately 250 genes involved in a wide mechanistic range of cell functions [115,116].

Nrf2 is a modular protein comprising 605 amino acids in humans; in addition, it possesses seven conserved domains in different functions to control Nrf2 transcriptional activity (Figure 4) [117,118]. These domains are homologous to a protein deriving from erythroid cells with CNC (cap‘n’collar) homology (ECH) named Nrf2-ECH (Neh) [117]. The Neh1 domain comprises conserved region CNC-bZIP, which plays an essential role as a transcription factor and in heterodimerization processes associated with other bZIP proteins such as musculoaponeurotic fibrosarcoma (sMAF) proteins; these proteins can be found in their MafF, MafG, and MafK forms, which can recognize antioxidant response elements (AREs) capable of activating gene transcription [119]. The Neh2 N-terminal domain has negative control over the Nrf2 activity, i.e., it mediates Nrf2 ubiquitination and degradation processes. Neh2 has two highly conserved peptide sequences, namely, ETGE (high affinity) and DLG (low affinity) degrons, which interact specifically with the transcription factor mediated by ECH-associated protein 1 (Keap1); consequently, they play a key role in proteasomal degradation processes [119,120].

The Neh3, Neh4, and Neh5 domains are involved in transcriptional activation processes, since they can bind to different transcriptional machinery components. The Neh3 domain is located in the C-terminal region. However, it is worth highlighting that removing 16 amino acids from the C-terminus of this protein inactivates the CNC-bZIP factor; this indicates the role played by it in target genes’ transactivation processes. Furthermore, the Neh3 domain can interact with Chromodomain Helicase DNA Binding Protein 6 (CHD6), a fact that corroborates its role in transcriptional activation processes [121,122].

The Neh4 and Neh5 domains act in a cooperative manner by interacting with cyclic adenosine 3′,5′-monophosphate (cAMP)-responsive binding protein (CREB) and by synergistically increasing gene transcription rates. In addition, these domains can bind to HMG-CoA reductase degradation protein 1 (HRD1) to mediate Nrf2 degradation [122,123].

Neh6 is another domain accounting for Nrf2-negative regulation as well as for its Keap1-independent regulation [123]. Like Neh2, the Neh6 domain presents two peptide degrons, namely, DSGIS and DSAPGS. These degrons are recognized by the β-transducin-repeat-containing protein (β-TrCP), which accounts for mediating Nrf2 degradation in cells under distress conditions. It is important to emphasize that degron DSGIS has a phosphorylation site for the glycogen synthase kinase-3 (GSK-3) enzyme, which increases β-TrCP’s ability to suppress Nrf2 when it is modified by GSK-3 [115,124].

Finally, Neh7 is the most recently described domain and has a region that is yet to be fully explained. This region interacts with retinoic receptor X α (RXRα) to suppress Nrf2 activity and to prevent co-activators’ recruitment to the Neh4 and Neh5 domains [125]. Overall, these domains act in Nrf2 stability modulation as well as in the transcriptional activation of its target genes, such as transcriptional, post-transcriptional and post-translational regulation [115,118].

**Figure 4 pharmaceuticals-17-00232-f004:**
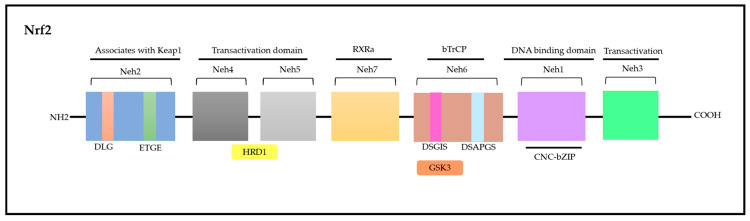
Nrf2 molecule—seven domains with various functions that control transcriptional activity. Adapted from Itoh [117] and Canning [122]. Nrf2: nuclear factor erythroid-derived 2; CNC: cap‘n’collar; ECH: erythroid cells with CNC homology; Neh: Nrf2-ECH; Keap1: ECH-associated protein 1; DLG: polypeptide sequence containing the amino acids aspartic acid, leucine and glycine; ETGE: polypeptide sequence containing the amino acids glutamate, threonine, glycine and glutamate; HRD1: HMG-CoA reductase degradation protein 1; RXRα: retinoic receptor X α; DSGIS and DSAPGS: peptidic degrons; GSK3: glycogen synthase kinase-3 β enzyme; bzip: Basic Leucine Zipper Domain; bTrCP: β-transducin-repeat-containing protein; DNA: desoxyribonucleic acid.

Overall, Nrf2 has a short half-life (approximately 15 min) under homeostatic conditions; further, it is linked to domains (Keap1 or β-TrCP or HRD1) capable of keeping it inactivated [9,126]. Its structure plays an essential role in helping us achieve a good understanding of its metabolic routes, since Nrf2 regulation mainly takes place through E3 ubiquitin ligase substrates involved in its ubiquitination and activation processes. These substrates comprise the Keap1-CUL3-RBX1 complex, SCF/β-TrCP, and HRD1 [113].

Each one of these complexes mediates Nrf2 degradation. In other words, they interrupt the connection with the above-listed domains to activate Nrf2, based on different stimuli; namely, Keap1-CUL3-RBX1 complex responds to electrophilic/oxidative modification of key cysteines, mTOR, and CUL3-Ring E3 ligase (CRL) inhibitors as well as to the competitive binding of ETGE-containing proteins and to increased p62/SQSTM1 levels, whereas SCF/β-TrCP can be modulated through metabolic changes taking place both in the cytosol and in cell nucleus (Figure 5). These changes are regulated by the glycogen synthase kinase-3 β enzyme, (GSK3β), by insulin or growth factors, and by CRL inhibitors. HRD1 ubiquitylates Nrf2 under endoplasmic reticulum stress [113].

However, it is important to emphasize that Nrf2 can also be regulated by other signaling pathways such as epigenetic (methylation, acetylation, and/or microRNAs) and post-translational factors (phosphorylation, ubiquitination, acetylation, and/or methylation) [8,113]. As previously mentioned, polyphenols, such as stilbenes, can activate Nrf2. Studies have shown that these phytochemicals often use signal transduction mechanisms involving a complex cascade of events that comprise the following phases: basal, pre-induction, induction, and post-induction [8]. After cell exposure to stilbenes, there is a pre-induction response via negative Nrf2 regulators, which are translocated from the cell nucleus to its cytoplasm. The induction phase takes place simultaneously to Nrf2 trans-location to the nucleus. This process is followed by stabilization and heterodimerization, which activate ARE-mediated cytoprotective gene expression and trigger the post-induction phase to interrupt Nrf2 activation [127,128].

When it comes to stilbenes, the most investigated pathway is the pathway acting through ARE; Nrf2 plays a key role in regulating antioxidant genes and phase-II metabolites. Heme oxygenase-1 (HO-1) and NAD(P)H:quinone acceptor oxidoreductase 1 (NQO1) were mostly identified among Nrf2 target genes [129,130]. The Nfr2/ARE pathway is suppressed under optimal conditions; Nrf2 remains trapped in the cytosol linked to the Keap1 domain that, together with cullin-3 (CUL3), forms the Keap1-CUL3-RBX1 complex, which is constantly exposed to ubiquitination and proteasomal degradation [131].

However, Nrf2 under stress conditions dissociates from the Keap1-CUL3-RBX1 complex through two pathways. The first relies on specific Keap1 cysteine residues’ modification by oxidants, phytochemicals (such as stilbenes), and/or electrophiles; whereas the second refers to specific Keap1 cysteine residues’ modification by p62 involved in autophagy process [131]. Both pathways enable Nrf2 translocation to the cell nucleus. Upon arriving in the cell nucleus, Nrf2 undergoes heterodimerization with small Maf proteins (sMaf) that play an essential role in its binding to ARE and, consequently, in the activation of target genes [4].

Alternatively, data have demonstrated that stilbenes can also act via new molecular targets (auxiliary mechanisms), such as via cAMP signaling, via AMP-activated protein kinase (AMPK), which accounts for regulating energy homeostasis, the estrogen-related receptor α (ERRα), and estrogen receptors (ER) as well as the enzymatic cofactor tetrahydrobiopterin (BH4)—an essential cofactor of the nitric oxide synthetase (NOS) enzyme, which accounts for nitric oxide (•NO) synthesis. Stilbenes can also act via phosphodiesterase (PDE, enzymes accountable for cAMP and cGMP degradation) mediated by increased cellular cAMP levels [132,133]. Therefore, the dimension surrounding stilbenes in the Nrf2 pathway activation process is quite complex, since it involves several factors accounting for activating specific target genes. This specificity means that stilbenes have therapeutic functions in some chronic and metabolic diseases. The latest evidence of stilbenes’ therapeutic effects via Nrf2 pathway mediation will be discussed below.

## 5. Stilbenes: Compounds-Based Approach

Although the literature reports several studies focused on investigating stilbenes, most of them remain at the preclinical testing stage, comprising models in vivo and in vitro, and show preference for RESV, a prototype of the class, over other compounds. The therapeutic action of other stilbenes, such as PIC, PIN, DHS, PTS, and PDT, has been recently explored. This section focuses on investigating each of the aforementioned stilbenes and their association with the mitigation of different diseases via Nrf2 nuclear pathway activation.

### 5.1. RESV

RESV (3,5,4′-trihydroxystilbene) is a natural polyphenol first identified in 1940 by Japanese scientists. RESV was found in the roots of *Veratrum grandiflorum* and, later, in the roots of *Polygonum cupsidatum*, the latter being an important traditional medicine in China [36,134,135,136,137]. RESV can be found in several plants, including peanuts (*Arachis hypogea*), blueberries and cranberries (*Vaccinium* spp.), Japanese knotweed (*Polygonum cuspidatum*)—a traditional local herbal medicine—and, even more widely and abundantly, in grapevine (*Vitis vinifera*) and its derivatives, such as red wines and whole grape juices [135,138,139].

Regarding RESV’s ability to promote health benefits through its role in activating Nrf2, several experimental studies have been conducted in this context; the vast majority of these studies were carried out in vivo (Table 1) and in vitro (Table 2), but some randomized clinical trials were also reported (Table 3) [140,141,142,143,144,145,146,147,148,149,150,151,152,153,154,155,156,157,158,159,160,161,162,163,164,165,166,167,168,169,170,171,172,173,174,175,176,177,178,179,180,181,182,183,184,185,186,187,188,189,190,191,192,193,194,195,196,197,198,199,200,201,202,203,204,205,206,207,208,209,210,211,212,213,214,215,216,217,218,219,220,221,222,223,224,225,226,227,228,229,230,231,232].

In view of the above, it is possible to note that the literature contains several intervention studies carried out with the administration of RESV in numerous contexts. Among them, most have been carried out on animal models, followed by combinations with or without cellular models; little research has been carried out on humans. When it comes to animal studies, most research has included rats and mice. However, experiments on birds and fish were also considered. RESV was administered orally, by gavage or intraperitoneally. The doses used varied in increasing concentrations and the administration periods ranged from a single dose to administration over a few days or months. The conditions tested were wide and varied, including bone, joint, kidney, cardiovascular, and neurodegenerative diseases, cancer, ageing, and wound healing. Although the investigations included several conditions, the main findings were uniform in observing the role of RESV in increasing the expression and/or signaling pathways involving Nrf2 as well as the antioxidant enzymes regulated by it. Furthermore, the action of this compound is also seen in negatively regulating NF-κB, attenuating inflammatory processes. It is also worth highlighting that the findings were independent of the dose and time of administration, emphasizing the importance of this antioxidant in situations involving oxidative stress and inflammation in animal models. Furthermore, in most cases among the studies analyzed, the administration of RESV was shown to alleviate clinical symptoms of the diseases, including improvement in insulin resistance, healing, as well as improvement in the lipid profile.

Like animal research, cellular model studies cover a wide range of varied conditions or diseases. The intervention time ranged from 4 to 72 h, with increasing concentrations, and the findings are unanimous in highlighting the important role of RESV in attenuating oxidative stress, increasing the expression of antioxidants, and/or promoting Nrf2 activation directly or indirectly through lower expression of inflammatory components, via NF-κB, as observed in animal models. In turn, very few clinical studies in humans have been carried out regarding the action of RESV on the activation of Nrf2, and only two are presented in the present study. Although both are of high methodological quality, randomized, double blind, and crossover, the findings are conflicting and underline the need for more research to be conducted from this perspective, aiming to analyze whether the promising results observed in animal models and in vitro can be confirmed in human populations. This strategy is attractive and capable of helping in the prevention and/or treatment of various adverse health situations. Furthermore, it is necessary to define better doses and administration times for the antioxidant for each condition evaluated to obtain more effective results and better determine RESV’s clinical applicability.

### 5.2. PTS

The first reports about PTS (trans-3,5-dimethoxy-4-hydroxystilbene) occurred in the 1940s when it was identified as a polyphenol in the bark of *Pterocarpus marsupium* (a deciduous tree); after a year, it was recognized and validated via synthesis. Subsequently, it was found in grapes, blueberries, and peanuts [232,233,234,235,236]. Its discovery redirected scientific interest, since previous attention regarding the use of stilbenes in health was centered on RESV [237].

PTS is a dimethylated analogue of RESV, which strongly affects its lipophilicity, increasing its availability in biological media, making it a more potent therapeutic agent than RESV [36,238,239]. Among its pharmacological properties, its antidiabetic property was the first to be ratified, through Ayurvedic medicine; however, amid scientific and technological advances in the area of phytotherapy applied to human health, other therapeutic activities were also found against numerous health problems and diseases [240].

In parallel with these findings, and sensitized by the discovery that pathological processes, in most cases, are initiated or intensified through the exacerbation of oxidative stress and inflammation, new scientific insights have emerged focusing on products that activate or suppress nuclear factors (Nrf2 and NF-κB), which modulate serum levels of ROS and proinflammatory cytokines. Given this, mechanistic studies, which analyze the pharmacokinetics of phytoalexins, identified that the biological capacity to attenuate the pathological processes of PTS mainly came from its ability to activate the Nrf2 pathway. Table 4 and Table 5 bring together the experimental studies in which PTS presented health benefits, via activation of Nrf2, with studies carried out in vitro and in vivo, respectively [241,242,243,244,245,246,247,248,249,250,251,252,253,254,255,256,257,258,259,260,261,262,263,264,265,266,267,268].

Given these findings, mostly in vivo studies could be identified (72.5%; 21/29 articles). The cellular viability of PTS was investigated in the concentration (dose) range of 2–50 µM, proving to be safe. However, it is noteworthy that depending on the cell culture, the tolerance threshold dose may be changed. For the animal model, doses ranging from 5 to 100 mg/kg were used, depending on the animal model used (rodents or zebrafish). It is notable that the activation of the Nrf2 pathway in the animal model was dependent on the PTS dose.

As tolerance data might vary as a result of the animal model, the use of several animal models may present a challenge when comparing results. Since the zebrafish model is a more modern animal model than the regularly used rats and/or mice, it has received less attention in the scientific literature while being regarded as a safe alternative model because of its genetic resemblance to humans.

### 5.3. PIC

PIC (3,3′,4,5′-*trans*-tetrahydroxystilbene) belongs to the stilbenes class and is an hydroxilated analogue of RESV; it was reported for the first time in 1956, isolated from the *Vouacapoua americana* species, also known as acapu, and is widely found in natural sources such as fruits, vegetables, and medicinal plants [269,270]. Studies report the presence of PIC in grapes, passion fruit, blueberries, white tea, and rhubarb [271].

Regarding the potential of this stilbene, it has demonstrated several biological activities such as antioxidant, antiviral, anticancer, antiglycant, antidiabetic, and anti-inflammatory activities [139,272,273,274,275]. Among its biological activities, modulation via Nrf2 was mainly found in animal models (Table 6) and in vitro (Table 7) [276,277,278,279,280,281,282,283,284,285]. It was observed that PIC positively regulated the expression of Nrf2 and its mRNA, increasing the expressions of NQO1, HO-1, γGCS, and GPx and, via Nrf2/Keap1, acting in testicular protection and attenuation of oxidative stress. In addition, a decrease in MDA and an increase in antioxidant enzymes SOD, CAT, and GPx were observed. The pathway observed in this study was Nrf2/Keap1, in which Nrf2 uncouples from Keap1, translocating Nrf2 to the cell nucleus, thus regulating target genes to promote an antioxidant response [279].

The concentration range of 5–20 mg/kg of PIC was found to be safe for use in animal model studies provided that the maximum dosage of 20 mg/kg is not exceeded. A concentration of 10 μM is utilized in most of the works pertaining to in vitro research. The highest concentration used among the studies reported was 40 μM. Nevertheless, in the current work, cell viability was assessed using concentrations as high as 40 μM, and it was found that PIC did not significantly affect cell viability at the concentrations tested [285].

Six hours to twelve weeks were employed for the evaluation of PIC intervention times, with 24 h being the most common intervention time in experimental methods. Kil et al. [274] found that, despite a shorter intervention period than prior research, there was a similar drop in ROS and rise in HO-1.

We observed that all studies have the same purpose, i.e., the activation of the Nrf2 pathway. The majority of studies using PIC as an activator of the Nrf2 pathway identified a common antioxidant enzyme, HO-1, which is important and plays an important role in protecting against oxidative injuries, modulating inflammation, regulating apoptosis, and contributing to angiogenesis [275]. Some studies evaluated greater activation pathways than others, depending on the analyzed conditions/diseases. Among the studies listed in Table 7, Achy-Brou et al. [283] evaluated the activation of Nrf2 through the reduction in ^●^NO levels, differing from other researchers who generally evaluated the production of antioxidant enzymes related to the activation of the Nrf2 pathway.

### 5.4. PIN and DHS

PIN (3,5-dihydroxy-trans-stilbene) is found in the Pinaceae family, mainly in the heartwood of *Pinus sylvestris* (also known as Scots pine), and can also be found in the leaves of *Pinus densiflora* [286,287]. Pine tree parts are traditionally used in East Asia to treat a variety of health conditions, including inflammation, liver toxicity, and stomach disorders. This compound is being studied extensively because it is very significant in plants and due to its positive effects on human health, including antioxidant, neuroprotective, and antiallergic properties [288]. Other activities reported for this compound are described in the literature, such as antibacterial [289] and anticancer activity [290]. Erasalo et al. [291] observed the anti-inflammatory effect in in vitro and in vivo models, inhibiting the PI3K/Akt signaling pathway.

DHS (4,4′-dihydroxy-trans-stilbene) can be found in the methanolic extract of *Yucca periculosa* bark [292]. Despite being less investigated, antioxidant and anticancer activity can be included among its biological properties [293,294]. According to Chen et al. [295], DHS was effective against pancreatic, ovarian, and colorectal cancer cells. This compound also showed an antimetastatic effect in vivo in a model of melanoma-mediated lung metastasis, where it was observed that DHS reduced the formation of large melanoma nodules [296,297].

To prevent information duplication, PIN and DHS were presented together in this section; some articles that studied their actions via Nrf2 also evaluated their pharmacokinetic features in a comparable manner. It is also valuable to note that, in comparison to the others, there are not many publications concerning these compounds in the scientific literature, perhaps because they were discovered more recently. Thus, Table 8 shows the results of in vitro tests, and Table 9 lists the in vivo experimental studies (in animal models) [298,299,300].

Wang et al. [300] evaluated the activation of the Nrf2 pathway in oligoasthenospermia for three stilbenes: PIN, DHS, and REVS. The dose tested in this study was 100 mg/kg, a much higher level compared to other studies carried out. Their research also evaluated changes in antioxidant enzymes, and a reduction in oxidative stress in the studied disease was observed.

There were two in vitro studies of PIN and DHS, both treating different conditions or diseases. PIN was used to treat oxidative stress-induced cell death, while DHS was used to treat COPD. Both studies shared a 24 h evaluation period and a concentration range very close to PIN (5 μM) and DHS (0.5–4 μM). PIN at a concentration of 5 μM elevated HO-1 levels, whereas Nrf2 levels remained basal in the study of PIN against oxidative stress induced in human retinal epithelial cells. The authors evaluated the role of p62, investigating the expression level of p62 mRNA in protection by PIN. They suggested that this may have occurred due to accumulation of the p62 protein [298].

### 5.5. PDT

PDT (3,4′,5-trihydroxystilbene-3-β-D-glucoside) is a glycoside derivative of RESV [301,302]. Like PTS, it has more prominent biological activity than RESV, particularly as an antioxidant [303]. It can be obtained from the Vitaceae, Liliaceae, and Leguminosae families; however, *Polygonum cuspidatum* and *Reynoutria japonica* are the main sources for extraction on an industrial scale [304,305].

The therapeutic and protective effects of PDT have been widely investigated, mainly regarding its role in modulating nuclear factors, such as Nrf2, in order to improve antioxidant defense against pathological processes [302,306,307]. In view of this, Table 10 lists studies that focused on the pharmacokinetic performance of polidatin in various diseases or pathological conditions such as liver disease, inflammatory bowel disease, neurodegeneration, endometriosis, diabetes mellitus, kidney disease, ophthalmological conditions, acute myocardial infarction, osteoarthritis, lung disease, auditory disease, asthma, and allergy [307,308,309,310,311,312,313,314,315,316,317,318,319,320,321,322,323,324,325,326,327].

All the publications examined PDT’s therapeutic potential using animal models (rats, mice, and/or guinea pigs) and doses ranging from 7.5 to 500 mg/kg. Some of the research found a direct correlation between PDT’s medicinal effects and dosage. Due to the fact that the treatment time varied based on the dose administered, a dose discrepancy can also be seen, making a comparative examination of its effects problematic. While PDT can indeed act to increase the expression of Nrf2 and its antioxidant enzymes, it is unclear how this leads to an improvement in the serum levels of endogenous markers. Therefore, studies that standardize the dose and duration of intervention are required to investigate the long-term effects of PDT in order to provide scientific evidence to conduct more rigorous and well-designed studies in humans.

## 6. Conclusions and Future Perspectives

Based on the present review, it is evident that stilbenes could activate Nrf2 either directly or indirectly. They achieve this by affecting NF-κB and utilizing distinct biological pathways. This activation has implications for treating diseases or health conditions that involve inflammation and redox imbalance. Nevertheless, most documented experimental experiments were carried out using cellular or animal models. Although the positive outcomes are evident, it is crucial to conduct multiple studies that adopt a multidisciplinary approach to thoroughly investigate the role of stilbenes in the Nrf2 pathway and their potential therapeutic uses. This entails gaining a comprehensive understanding of the biology and biochemistry of how these compounds interact with other molecules, developing other laboratorial methods (enzyme interaction investigations), conducting dose–response studies, and performing preclinical and clinical controlled trials, as well as searching for and guaranteeing the quality, purity, and stability of the chemical compounds or extracts used. These investigations are of utmost importance in exploring the therapeutic properties of natural products and their behavior in pathophysiological conditions. Consequently, investigations conducted on cellular and animal models play a vital role in assessing the safety and effectiveness of herbal medicines. These studies pave the way for further research involving human subjects and ultimately lead to the development of new formulations that can enhance the well-being and quality of life for individuals afflicted with health conditions that pose a threat to their overall health. In addition, it is crucial to consider the potential interaction of stilbenes with other metabolic or physiological processes. This contact could result in broader impacts across the body when these chemicals are used. Therefore, it is imperative to further explore this aspect. Furthermore, it is important to emphasize the essential significance of stereochemistry in the profound modification of the reactivity and biological activity of stilbenes. Therefore, this research specifically examines the arrangement of *E*-stilbene isomers and their hybrid derivatives, since they possess greater stability, lower cytotoxicity, and more significant biological activity as compared to the Z-stilbenes configuration. There is a deficiency in scientific and technological expertise. To ensure optimal activation of Nrf2 and/or negative regulation of NF-κB, it is crucial to establish precise dosages, delivery methods, and timing for each type of stilbene and for each specific situation. This will enable the standardization of supplementation with each chemical. Furthermore, it is imperative to assess the safety of stilbenes in the context of different situations where their potential benefits have been proposed, with the goal of offering more reliable and secure data.

## Figures and Tables

**Figure 1 pharmaceuticals-17-00232-f001:**
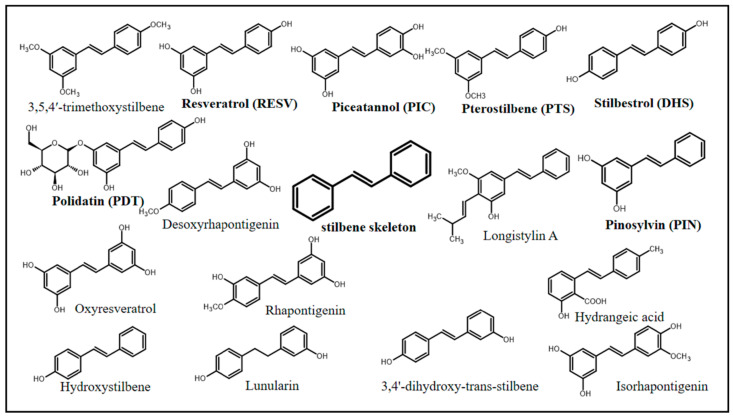
Chemical structures of stilbenes with documented biological activity.

**Figure 3 pharmaceuticals-17-00232-f003:**
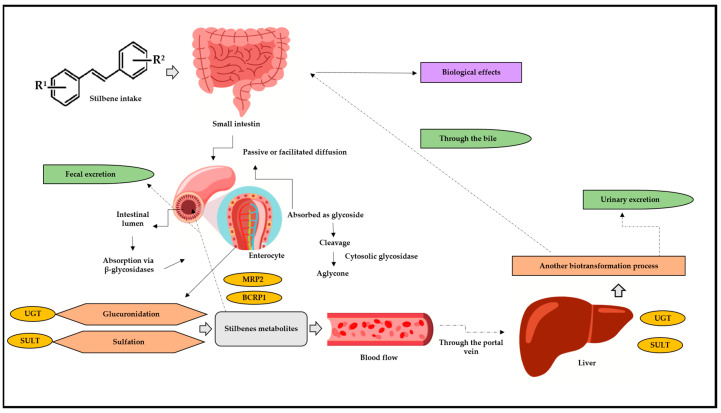
Stilbenes’ biotransformation in human body. BCRP1: breast cancer resistance protein 1; MRP2: multidrug resistance protein 2; UGT: glucuronosyl-transferase; SULT: sulfotransferase; R1: substituent 1; R2: substituent 2. After oral or intragastric intake, stilbenes reach the small intestine, where they can be absorbed via passive or facilitated diffusion. In passive diffusion, stilbenes must be absorbed as a glycoside and then cleaved, generating aglycone through the action of the cytosolic glycosidase enzyme. In turn, this process can occur through the action of β-glucosidases in the intestinal lumen. Upon reaching the enterocytes, stilbene is metabolized by glucuronidation reactions, through the action of UGT, and via SULT, through sulfotransferase. After these biotransformation steps, the metabolites return to the intestinal lumen through efflux and action of the enzymes BCRP1 and MRP2. Arriving in the bloodstream, these metabolites go through the portal vein and straight to the liver, where they undergo a new biotransformation process. Then, they can return to the small intestine through the bile, be absorbed, and, finally, exert their biological effects and/or can be excreted through urine.

**Figure 5 pharmaceuticals-17-00232-f005:**
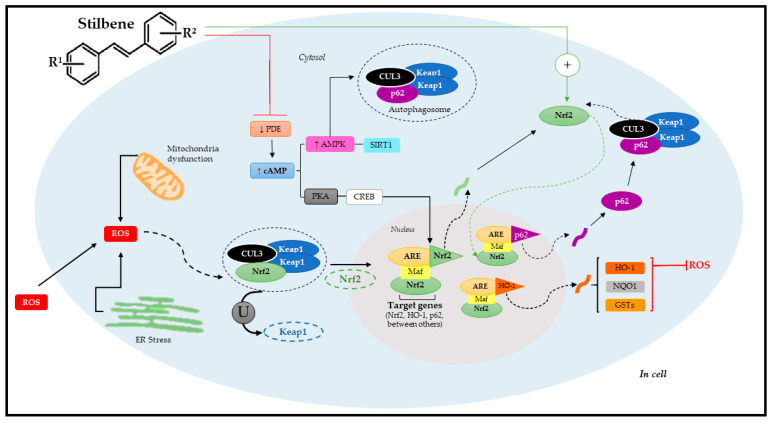
Molecular mechanism of Nrf2 pathway activation through stilbenes. Adapted from Reinisalo [4] and Niture [8]. R^1^: substituent 1; R^2^: substituent 2; ROS: reactive oxygen species; CUL3: cullin-3; Nrf2: nuclear factor erythroid-derived 2; ER: endoplasmic reticulum; ARE: antioxidant response elements; Keap1: ECH-associated protein 1; U: ubiquitination; PDE: phosphodiesterase; cAMP: cyclic adenosine 3′,5′-monophosphate; PKA: protein kinase A; CREB: cAMP-responsive binding protein; AMPK; SIRT1: sirtuin 1; p62: autophagy substrate; HO-1: Heme oxygenase-1; NQO1: NAD(P)H:quinone acceptor oxidoreductase 1; GSTs: glutathione S-transferases; Maf: musculoaponeurotic fibrosarcoma proteins.

**Table 1 pharmaceuticals-17-00232-t001:** List of resveratrol (RESV) uses and Nrf2 pathway-based activity in animal models.

Author and Year	Sample (*n*)	Route of Administration	Intervention Time	Doses	Condition/ Disease	Nrf2 Pathway
Kim et al. [140]	7 mice/ group	Oral	6 months	I: 40 mg/kg	Progressive kidney damage caused by aging	RESV improved proteinuria, histological changes, inflammation and ↑ NRF2Nrf2 expression, improving oxidative stress and mitochondrial dysfunction.
Bhattarai et al. [141]	5 rats/ group	Subcutaneous and palatal gum injection	2 weeks	I: RESV and DMSO groups: 5 mg/kg body weight daily	Alveolar bone loss	RESV almost completely inhibited the alterations promoted by the oxidative stress ↑ HO-1, mediated by NRF2.
Wang et al. [142]	6 rats/ group	Intradermal injection	24 days	I: 10 mg/kg	Rheumatoid arthritis	RESV ↓ AA scores and serum levels of antioxidant enzymes, and inhibited ROS production by activating the SIRT1/Nrf2 signaling pathway.
Kong et al. [143]	10 mice/ group	Gavage	15 days	I: 40 mg/kg or 20 mg/kg)	Alzheimer’s disease	RESV ↑ the antioxidant capacity of animals in the experimental group, such as SOD, CAT, GPx, through the NRF-2/HO-1 signaling pathway.
Cui et al. [144]	10 mice/ group	Gavage	6 weeks	I: 2 mg/kg	Photoaging	A protective effect was seen against photoaging though the ↓ in the expression of matrix metalloproteinases and inflammatory factors, inhibiting the ROS production measured by the MAPK and COX-2 pathways, in addition to promoting the NRF2 signaling pathway.
Zhou et al. [145]	6 rats/ group	Subcutaneous injection	2 weeks	C: rats’ cutaneous wounds were only washed daily with physiological saline I: rats’ cutaneous wounds were washed daily with physiological saline followed by RESV local application.	Wound healing	RESV ↑ Nrf2 and Mn-SOD, and subsequently attenuated oxidative stress, promoting the acceleration and quality of healing of cutaneous wounds.
Xun et al. [146]	6 piglets/ group	Intraperitoneal injection	21 days	I: 10, 30 or 90 mg/kg	Induced intestinal integrity and inflammation	RESV protects intestinal integrity, alleviates intestinal inflammation and oxidative stress by modulating AhR/Nrf2 pathways in piglets challenged with diquat (organic herbicide).
Hosseini et al. [147]	10 mice/ group	Oral	16 weeks	C: Standard chow diet (10 kcal% fat) I_1_: a high fat diet (HFD, 55.9 kcal% fat) I_2_: HFD-supplemented with 0.4% HFD + RESV groups	Non-alcoholic fatty liver disease	RESV attenuated HFD induced methylation of the Nrf2 promoter in the liver of mice, and this effect was correlated with ↓ in triglyceride levels and ↓ in expression of genes related to lipogenesis, such as FAS and SREBP-1c.
Zhao et al. [148]	8 rats/ Group	No description	7 days	I: 50 mg/kg for 7 days	Acute respiratory distress induced by seawater inhalation	AC-Res attenuated respiratory distress via Trx-1 and Nrf2, both in animals.
Yang et al. [149]	10 ducks/ group	Oral	15 days	I: the basal diet supplemented with 400 mg/kg RESV	Birds subjected to heat stress	RESV significantly activated the SIRT1-NRF1/NRF2 signaling pathways, and ↑ SOD and CAT. NF-κB/NLRP3 inflammasome signaling pathways were repressed under acute heat stress. Meanwhile, RESV supplement further inhibited the NLRP3 inflammasome pathway.
Rasheed et al. [150]	No description	Intraperitoneal	9 weeks	I: 10 mg/kg	Parkinson’s disease	RESV promotes the catalytic activity of the xenobiotic-metabolizing enzyme, Cyp2d22/CYP2D6, which partially contributes to Nrf2 activation in pesticide-induced parkinsonism.
Recalde et al. [151]	16 rats/ group	Intraperitoneal injection	21 days	I: (RESV ethanol 0.1%) in their drinking water	Hemotherapy-induced peripheral neuropathic pain (cancer)	RESV prevented the upregulation of NFκB, TNF-α, ATF3 and c-fos, while ↑ expression of Nrf2, NQO-1, HO-1 and the redox-sensitive deacetylase SIRT1. RESV treatment was also able to restore TBARS levels and the GSH/GSSG ratio.
Wang et al. [152]	18 mice/ group	Oral	Until 6 months	I: 10 mg/kg/day for 1 month. After another month, a subset of mice was used for experimentation; the remaining mice were maintained for another 2 or 5 months (corresponding to 3 or 6 months after RESV treatment) and then were used for experimentation	Diabetic cardiomyopathy	RESV prevents DM-induced cardiomyopathy, in part, by ↑ Nrf2 expression and transcriptional activity.
Wang et al. [153]	7 mice/ group	Intragastric administration	No description	I: 5, 10, 20 mg/kg	Allergy	RESV exerts an inhibitory effect on MRGPRX2-mediated mast cell activation by targeting the Nrf2 pathway, and may present a promising new therapeutic agent for allergy, avoiding anaphylactoid reactions.
Trusov et al. [154]	No description	Oral	No description	C: standard balanced diet I_1_: high-fat-high-carbohydrate diet (HFCD) with an excess of total fat (30%) and fructose (20% solution instead of drinking water) I_2_: the same diets supplemented with RESV in a low (25 mg/kg body weight as RESV and 300 mg/kg as L-carnitine) or high (50 and 600 mg/kg body weight, respectively) doses	Obesity	RESV combined with L-carnitine caused ↓ in the number of positive cells for Nrf2 and ICAM-2 in the liver of rats treated with diets with high concentrations of carbohydrate and fat, but had the opposite effect on the kidneys. RESV + L-carnitine at a low dose by the same group caused alterations in the expression profiles of the studied marker genes, indicating a possible hypolipidemic effect.
Wang et al. [155]	12 rats/ group	Intraperitoneal injection	18 h	I: 30 mg/kg was administered at 6 h after surgery and then again at 12 h	Pediatric acute kidney injury	In the RESV group, the induced kidney injury was alleviated by ↓ the expression of TNF-α, IL-1β and KIM-1. Nrf2 signaling is known to effectively inhibit inflammation. Further, there were ↑ in the expression of HO-1 and NQO1in the RESV group.
Wu et al. [156]	15 mouse/ group	Gavage	Alternate days for 2 weeks	I_1_: 30 mg/kg I_2_: 100 mg/kg	Ovarian aging	RESV alleviated the loss of oogonial stem cells and showed a mitigating effect on induced oxidative apoptosis in mouse ovaries, which can be attributed to attenuation of oxidative levels in the ovaries activating Nrf2. Further, ↑ SIRT1 and FOXO1 and ↓ NF-κB.
Sun et al. [157]	6 rats/ group	Gavage	5 weeks	I: 30 mg/kg	Induced myocardial injury	RESV attenuated the oxidative stress through the expression of antioxidant molecules via Nrf2. Furthermore, AMPK may play a role in Nrf2/HO-1 signaling by RESV.
Kabel et al. [158]	10 rats/ group	Gavage	24 weeks	I: 30 mg/kg	Renal carcinoma	Intervention induced improvement in renal functions with a significant ↑ in tissue antioxidant defenses and Nrf2/HO-1 content associated with a significant ↓ in TGF-β1, TNF-α, IL-6 and STAT3 and alleviated histopathological and immunohistochemical changes compared to the untreated renal carcinoma group.
Lian et al. [159]	10 rats/ group	Oral	12 weeks	I: 50 mg kg	Obstructive sleep apnea associated with lung injury	Nrf2 and HO-1 protein levels were ↓ in the chronic intermittent hypoxia group compared to the control group. Rats in the RESV group had ↓ percentages of apoptotic cells, levels of IL-6, TNF-α, Bax, and cleaved caspase-3, and ↑ levels of Nrf2 protein and HO-1.
Yang et al. [160]	60 ducks/ group	Oral	15 days	I: basal diet supplemented with 400 mg/kg RESV	Growth performance and anti-inflammatory ability	Dietary RESV can improve growth performance and ↓ inflammation through Nrf2/HO-1 and TLR4/NF-κB signaling pathways in ducks.
Zhou et al. [161]	8 rats/ group	Intragastric	18 weeks	I: 50 mg/kg every other day	Breast cancer	RESV treatment could upregulate the expression of Nrf22 and UGT1A8, accelerate metabolic elimination of catechol estrogens, inhibit estrogen-induced DNA damage and suppress the pathological development of breast cancer.
Xu et al. [162]	10 rats/ group	Intraperitoneal injection	7 days	I: 20 mg/kg	Type 2 diabetes	RESV ↑ Nrf2 expression in a diabetic heart by stimulating SIRT1 or inhibiting GSK3β, alleviating myocardial oxidative stress and improving ischemia–reperfusion injury.
Izquierdo et al. [163]	Until 20 mice/ group	Oral	2 months	I: 1 g/kg	Senescence	Maternal RESV supplementation may prevent cognitive impairment in mouse offspring through epigenetic alterations and Nrf2 signaling pathways.
Chen et al. [164]	12 piglets/ group	Intraperitoneal injection	15 days	I: 300 mg/kg	Redox status and intestinal microbiota	RESV and PTS administration ↑ jejunal SOD activity and SOD 2 mRNA and protein expression of IUGR piglets, promoting Nrf2 nuclear translocation. PTS was superior to RESV in ↑ Nrf2 nuclear translocation and inhibiting MDA accumulation in the jejunum of IUGR piglets. Further, RESV modulated the composition of the fecal microbiota of IUGR piglets.
Zeng et al. [165]	10 mice/ group	Intragastric	32 days	I: 10 to 100 mg/kg	Methamphetamine-induced memory deficit	Pretreatment with RESV in methamphetamine-induced memory dysfunction was possibly related to activation of the Keap1-Nrf2 pathway and ↓ of apoptosis, suggesting benefits of using this antioxidant in this condition.
Cong et al. [166]	15 mice/ group	Intraperitoneal injection	10 days	I_1_: 25 mg/kg I_2_: 50 mg/kg	Traumatic brain injury	RESV has a protective effect on brain injury induced by chest blast exposure, likely mediated by Nrf2/Keap1 and NF-κB signaling pathways.
Javkhedkar et al. [167]	8–12 rats/ group	Oral	9 weeks	I: 50 mg/kg	Renal interstitial inflammation in the pathogenesis of hypertension	Long-term RESV administration restores Nrf2 expression, improves inflammation and attenuates the development of hypertension.
Rubio-Ruiz et al. [168]	16 rats/ group	Oral	20 weeks	I: RESV and quercetin 50–0.95 mg/kg/day, respectively	Fatty liver in metabolic syndrome	RESV + quercetin has beneficial effects on oxidative stress in fatty liver of rats with metabolic syndrome through the improvement of antioxidant capacity and overexpression of the Nrf2 factor, which increases enzymes antioxidants and GSH recycling.
Chen et al. [169]	5–6 mice/ group	Intragastric	13 days	I: 200 mg/kg	Colitis and colon cancer	The RESV analogue studied has stronger anticolitis effects than RESV, with even greater ability to stimulate the Nrf2 pathway than the parent compound.
Pierre et al. [170]	16–17 mice/ group	Oral	8 weeks	I: 1–10 mg/kg	Erectile Dysfunction	RESV and MitoQ were ineffective in reversing the effects of androgen deprivation on vascular reactivity; however, treatment with high doses of RESV upregulated several important antioxidant genes including CAT, SOD1, GSTm1, Prdx3, and Nrf2.
Kim et al. [140]	8 mice/ group	Oral	2 weeks	I: 2 mg/kg	Photoaging	Nrf2-dependent antioxidant enzymes induced by grape skin extract or RESV, including HO-1 in the liver and skin, as well as inhibited metalloproteinases and attenuated UVB-induced photoaging through activation of the Nrf2/HO-1 signaling pathway.
Wang et al. [171]	9 birds/ group	Oral	21 days	I: 400 mg/kg	Intestinal growth and development	RESV may improve intestinal development and antioxidant function in broiler chickens under heat stress. Further, the RESV group shows ↑activities of GPX, GST and mRNA levels of Nrf2 and SOD1.
Ma et al. [172]	10 mice/ group	Intraperitoneal injection	5 days	I: 25 mg/kg	Diabetic cardiomyopathy	RESV actives SIRT1, induces ↑ expression of Nrf1 and Nrf2 and ameliorates diabetic cardiomyopathy.
Wang et al. [173]	7–8 mice/ group	Oral	4 months	I: 30 mg/kg	Cognitive impairment caused by type 2 DM	RESV prevented the cognitive impairment induced by DM2 through anti-inflammatory and antioxidant activities. This effect was accompanied by the upregulation of Nrf2 transcriptional activity and ↑ expression of antioxidant genes.
Ikeda et al. [174]	No description	Intraperitoneal injection	on day 15 after periodontitis induction	I: 10 mg/kg	Periodontitis	RESV dimer induced greater periodontal bone healing when compared to the use of RESV monomer. It appears that periodontal bone healing in both groups was likely related to master regulation of Nrf2 and downregulation of IL-1β in the RESV dimer group.
Cirmi et al. [175]	7 mice/ group	Oral or Intraperitoneal injection	14 days	I: 20 mg/kg	Cadmium-induced kidney injury	Cadmium caused damage to glomeruli and tubules and ↑ expression of Nrf2 genes. RESV significantly improved all parameters.
Jia et al. [176]	10 mice/ group	Intraperitoneal injection	6 days	I: Oxyresveratrol 40–80 mg/kg	Acute liver injury	Hepatoprotective efficacy of oxy-resveratrol was seen due to the prevention of TLR4/NF-κB pathway activation, induced activation of the Keap1-Nrf2 signaling pathway, and decreased hepatocyte apoptosis.
Seo et al. [177]	5 mice/ group	Intraperitoneal injection	8 days	I: 20 mg/kg	Atherosclerosis	RESV inhibits ICAM-1 expression via transcriptional regulation of FERM-kinase and Nrf2 interaction, thus blocking monocyte adhesion, suggesting that RESV improves inflammation and delays the onset of atherosclerosis.
Zhang et al. [178]	3 mice/ group	Intragastric	12 weeks	I: 10 mL/kg 10%	Diabetic neuropathy	RESV may attenuate the severity of diabetic neuropathy by protecting peripheral nerves from apoptosis, inhibiting the NF-κB pathway, and ↑ Nrf2 expression.
Krajka-Kuźniak et al. [179]	3 mice/ group	Topical use	3 days	I_1_: 8 µM/kg I_2_: 16 µM/kg	Mouse epidermis	RESV and its methylthioderivatives activate Nrf2 in mouse epidermis and promotes upregulation of GST.
Zhou et al. [180]	6 mice/ group	Gavage	Day 14 of pregnancy until parturition	I: 50 mg/kg	*Streptococcus uberis* infection	RESV can function as an activator of the p62–Keap1/Nrf2 signaling pathway to improve oxidative injury caused by S. uberis in mammary glands as well as in EpH4-Ev cells.
Cheng et al. [181]	10 rats/ group	Intravenous	After ischemia induction, RESV was administered 5 min before reperfusion	I: 100 μmol/L	Myocardial ischemia	RESV exerted significant antioxidant and cardioprotective effects after myocardial ischemia, possibly through activation of the Nrf2/ARE signaling pathway.
El-Fattah et al. [182]	6–8 rats/ group	Oral	30 days before and 45 after the induction of testicular dysfunction	I: 80 mg/kg	Di-(2-ethylhexyl)phthalate-induced testicular dysfunction	Pretreatment with RESV and curcumin were able to recover the lesions induced in this model. The chemoprotective effects of these compounds may be due to their intrinsic antioxidant properties, along with increased gene expression levels of Nrf2, HSP 60, HSP 70 and HSP 90.
Yang et al. [149]	15 ducks/ group	Oral	22 days	I: 300, 400 or 500 mg/kg	Ileitis caused by LPS	RESV alleviated acute ileitis induced by duck LPS through Nrf2 and NF-κB signaling pathways, and dietary RESV 500 mg/kg is more efficient.
Muhammad et al. [183]	7 mice/ group	Oral	8 weeks	I: 200 mg/kg	Alzheimer’s disease	Induced memory deficit was improved by RESV. Both Aβ and Nrf2 ↑ significantly in the group that received RESV, with isolated treatment with this compound being the most effective.
Elbaz et al. [184]	8 rats/ group	Oral	7 days	I: 20 mg/kg	Diclofenac-induced hepatorenal toxicity	A potential therapeutic role for RESV in mitigating the hepatorenal insult induced by diclofenac has been demonstrated, possibly via modulation of the Nrf2/GSH axis.
Jia et al. [185]	30 animals/ group	Oral	60 days	I: 0, 0.1, 0.3, and 0.6 g/kg	Induced liver damage	RESV protection against H_2_O_2_-induced liver damage, inflammation and immunotoxicity was due to its antioxidant property and its ability to modulate Nrf2 and TLR2-Myd88-NF-κB signaling pathways.
Cheng et al. [186]	6 mice/ group	Oral	12 weeks	I: 10 mg/kg	Hyperglycemia and pancreatic damage	RESV treatment markedly improved the blood glucose level of the oral glucose tolerance test and promoted Nrf2 phosphorylation in the pancreas of mice treated with methylglyoxal.
Liu et al. [187]	90 ducks/ group	Gavage	70 days	I: 500 mg/kg	Acute liver damage induced by aflatoxins	RESV increased phase II enzyme activity, activate Nrf2 signaling pathway, and protect duck liver from toxicity, oxidative stress, and inflammatory reaction.
Wei et al. [188]	10 rats/ group	No description	8 weeks	I: 50 mg/kg	Osteoarthritis	RESV improves inflammatory damage and protects against osteoarthritis in a rat model via NF-κB and HO-1/Nrf-2 signaling.
Zhou et al. [189]	50 animals/ groups	Oral	7 days	I: 500 mg/kg	Diquat-induced intestinal oxidative stress	Dietary supplementation with RESV and apigenin attenuates oxidative stress involving NRF2 signaling pathways in diquat-challenged pullets.
Zhang et al. [190]	5 mice/ group	Intraperitoneal injection	7 days	I: 20 mg/kg	Osteogenic potential	RESV ↓ oxidative stress by alleviating proliferation, mitigating ROS activity, ↑ SOD enzyme activity and improving GSH concentration, been confirmed by gene expression of SOD1 and Sirt1/Nrf2.
Wu et al. [191]	53–54 animals/ group	No description	Until 40 days	I: 0.5 mM/mL	Cerebellar neurotoxicity in spinocerebellar ataxia type 3	RESV and caffeic acid were able to ↑ activation of Nrf2 in this model.
Hussein and Mahfouz [192]	6 rats/ group	Oral	8 weeks	I: 5 mg/kg	Diabetic nephropathy	RESV alone or co-administered with rosuvastatin improved antioxidant status back to control values. RESV with rosuvastatin remarkably normalizes the renal expression of TGF-β1, fibronectin, NF-κB/p65, Nrf2, Sirt1 and FoxO1 in the diabetic rat group.
Li et al. [193]	5 rats/ group	Intragastrical administration	24 h	I: RESV was diluted in sterile saline to 0.23 μg/kg, 1,5 mL	Renal ischemia–reperfusion injury (IRI)	RESV Shows renoprotection exerting significant effects on inflammatory responses, oxidative stress and apoptosis through the Nrf2/TLR4/NF-κB pathway.
Tamaki et al. [194]	6 rats/ group	Oral	3 weeks	I: 10 mg/kg	Periodontitis	RESV exhibited multiple beneficial effects in preventing periodontitis and ↓ oxidative stress through its activation of both Sirt1/AMPK and Nrf2/antioxidant.
Xu et al. [195]	10 rats/ group	Intraperitoneal injection	6 weeks	I: 20 mg/kg	Myocardial ischemia–reperfusion in diabetic rats	RESV can inhibit oxidative stress and alleviate MIR by activating the AMPK/p38/Nrf2 signaling pathway.
Gao et al. [196]	12 rats/ group	Intraperitoneal injection	7 days	I: 40 mg/kg	Hypoxic-ischemic encephalopathy (HIE)	RESV exhibited neurotherapeutic potential through upregulation of expression of Nrf2 and HO-1 signaling pathway proteins and thereby attenuate oxidative stress and inflammatory response.
Li et al. [197]	6 rats/ group	Oral	6 weeks	I: RESV supplement of 0.03%	Oxidative stress in obese asthmatic rats	RESV significantly ↑ CAT, GSH, GPx, and total SOD levels compared to obese, asthmatic, and untreated obese asthmatic rats. Furthermore, it significantly ↓ levels of Keap-1 and↑ Nrf2 in the heart, lung and kidney tissues of rats compared to untreated controls.
Li et al. [198]	6 mice/ group	Intraperitoneal injection	72 h	I: 30 mg/kg	Lung injury	RESV is a SIRT1 activator and strongly ↑ SIRT1 expression and attenuated lung injury. Furthermore, RESV treatment ↑ the expression of NRF2 and GSH, ↑ the activity of HO-1, SOD and CAT, but ↓ the MDA expression.
Singh et al. [199]	No description	Subcutaneous pellet	Every other month, until 8 months	I: 50 mg/kg	Breast cancer	RESV alone or in combination with 17β-estradiol significantly ↑ the expression of Nrf2 in breast tissues. Expression of antioxidant genes regulated by NRF2, NQO1, SOD3 and OGG1 that are involved in protection against oxidative DNA damage were ↑ in breast tissues treated with RESV and RESV + 17β-estradiol.
Xu et al. [200]	6 mice/ group	Intraperitoneal injection	7 days	I: 10 mg/kg	Acute liver toxicity	RESV/FGF1↓ oxidative stress and thereby alleviated liver injury by promoting nuclear translocation of Nrf2 and subsequently ↑ expression of antioxidant proteins in an AMPK-dependent.
Lu et al. [201]	6 mice/ group	Intraperitoneal injection	7 days	I: 10 mg/kg	Doxorubicin-induced cardiotoxicity	RESV could reduce the growth-promoting activity of FGF1. The co-treatment of RESV and FGF1 exhibits a more powerful cardio-antioxidative capacity in a treated model. The inhibition of SIRT1/NRF2 abolished RESV in combination with FGF1 on cardioprotective action.
Meng et al. [202]	20 animals/ group	Oral	20 days	I: 30 mg/kg	Pregnancy and lactation	In the placenta, Nrf2 protein expression was ↑ and Keap1 protein expression was ↓ by dietary RESV. mRNA expression of antioxidant genes including CAT, GPX1, GPX4, SOD1 and HO1.
Wang et al. [203]	10 animals/ group	Oral or intraperitoneal injection	28 days	I: 600 mg/kg	Ovarian oxidative stress	RESV reversed the tryptophan-kynurenine pathway, ↑ levels of Nrf2 and SIRT1, and ↓ FoxO1 and P53.

↑: increase; ↓: decrease; AC-Res: acetylresveratrol; AMPK: adenine monophosphate-activated protein kinase; ATF3: activating transcription factor 3; C33: imazosulfuron, pesticide/herbicide; CAT: catalase; C-fos: The cellular oncogene c-Fos; COX-2: cyclooxygenase 2; DM: diabetes mellitus; DMBA: 7,12-dimethylbenz(a)anthracene; FERM: kinase-ezrin, radixin, moesin homology domain; FGF1: Fibroblast growth factor 1; GSH/GSSG: oxidized glutathione/reduced glutathione ratio; GSK3β: glycogen synthase kinase 3β; GSTm1: glutathione s-transferase; HFD: high fat diet; HO-1: heme oxigenase-1; ICAM: intercellular adhesion molecule 1; IRA-2-methoxyl-3,6-dihydroxyl; LPS: lipopolysaccharide; MIR: myocardial ischemia–reperfusion; NF-κB: nuclear factor kappa b; NQO-1: NAD(P)H quinone-oxidoreductase-1; Nrf2: nuclear erythroid 2-related factor; OXA: oxaliplatin; Prdx3: peroxiredoxin 3; RESV: resveratrol; ROS: oxygen reactive species; SIRT1: sirtuin 1; SOD: superoxide dismutase; SREBP-1c: sterol regulatory element binding proteins; STAT-3: signal transducer and activator of transcription 3; Trx-1: thioredoxin; TBARS: thiobarbituric acid reactive species.

**Table 2 pharmaceuticals-17-00232-t002:** List of resveratrol (RESV) uses and Nrf2 pathway-based activity in vitro studies.

Author and Year	Cell Culture	Intervention Time	Concentration	Condition/Disease	Mechanism in Nrf2
Achy-Brou et al. [204]	Macrophages	Until 24 h	Increasing concentrations of RESV or PTS (0, 5, 10, 20 or 30 μM)	Macrophage stimulation by the danger signal LPS	LPS stimulation ↓ the cytotoxicity of RESV but not of PTS in these cells. RESV and PTS were each found to separately and significantly ↑ caspase 3 activity, which is related to NRF2. LPS stimulation prevented caspase 3 activation by PTS and ↓ caspase 3 activation by RESV in cells.
Bhattarai et al. [141]	Human gingival fibroblasts	48 h	0–200 μM	Alveolar bone loss	RESV almost completely inhibited the alterations promoted by the oxidative stress ↑ HO-1, mediated by NRF2.
Cui et al. [144]	Human epidermal keratin-forming cells	24 h	0, 10, 20, 40, 60, 80, and 100 μM	Photoaging	A protective effect of RESV was seen against photoaging in the cells studied due to the ↓ in the expression of matrix metalloproteinases and inflammatory factors, inhibiting the production of ROS measured by the MAPK and COX-2 pathways, in addition to promoting the Nrf2 signaling pathway.
Li et al. [205]	Microglial cells	24 h	50 μM	Inflammation and oxidative stress in microglial cells	RESV attenuated rotenone-induced inflammation and oxidative stress in this cells through ↑ inhibition of STAT1 and Keap1 and upregulation of Nrf2 and SLC7A11.
Hosoda et al. [206]	Myoblasts	4 h	No description	Antioxidant and antiapoptotic effect	Piceatannol and RESV were different in cytotoxicity, oxidant scavenging activities and cytoprotective mechanisms. The protection of piceatannol against ROS-induced apoptosis was superior to that of RESV. In addition to the SIRT-1-dependent pathway, PIC exerted HO-1-related Nrf2-mediated antioxidant and antiapoptotic effects, which could be an advantage of PIC compared to RESV.
Zhou et al. [145]	Human umbilical vein endothelial cells	24 h	100 nM	Wound healing	RESV ↑ Nrf2 and Mn-SOD, and subsequently attenuated oxidative stress, promoting the acceleration and quality of healing of cutaneous wounds.
Hosseini et al. [147]	HepG2 cells	48 h	10, 20, 25, 40, 50, 80, 100, and 200 μM	Non-alcoholic fatty liver disease	Treatment of HepG2 cells with high glucose ↑ the Nrf2 promoter methylation, while RESV reversed this effect. Treatment of cells with RESV can prevent high glucose-induced ROS production and the expression of antioxidant genes controlled by Nrf2. RESV attenuates NAFLD through epigenetic modification of Nrf2 signaling.
Chen et al. [207]	Human keratinocytes	24 h	0.1–1.0 μM	Protective effects against oxidative stress induced by 5-fluorouracil (chemotherapy) and inflammatory responses	RESV suppressed 5-FU-induced ROS overproduction, upregulating antioxidant defense genes through activation of Nrf2 and SIRT-1 expression. Regarding inflammatory responses, RESV suppressed 5-FU-induced expression of pro-inflammatory cytokines via nuclear translocation of NF-κB.
Chen et al. [164]	Jejunal enterocitos	25 h	0, 1, 2.5, 5, 10, 25, 50 μM	Redox status and intestinal microbiota	RESV and PTS protected against H_2_O_2_-induced mitochondrial dysfunction by facilitating mitochondrial biogenesis and ↑ the activities of mitochondrial complexes. In addition, both RESV and PTS efficiently mitigated mitochondrial oxidative stress by ↑ SIRT3 protein expression and the deacetylation of SOD2 and peroxiredoxin 3.
Zhao et al. [148]	Alveolar macrophage cell and alveolar epithelial cell	4 h	40 µg/mL	Acute respiratory distress induced by seawater inhalation	RESV attenuated respiratory distress via Trx-1 and Nrf2, at the cellular level.
Rasheed et al. [150]	Neuroblastoma cells	48 h	No description	Parkinson’s disease	RESV promotes the catalytic activity of the xenobiotic-metabolizing enzyme, Cyp2d22/CYP2D6, which partially contributes to Nrf2 activation in pesticide-induced parkinsonism.
Wang et al. [153]	Human mast cells	24 h	0, 6.25, 12.5, 25, 50, 100, 200, 400, 800 μM	Allergy	RESV exerts an inhibitory effect on MRGPRX2-mediated mast cell activation by targeting the Nrf2 pathway.
Zhang et al. [208]	Vascular smooth muscle cells	Until 72 h	5 mM or 10 mM	Vascular calcification	RESV may improve oxidative injury of vascular smooth muscle cells by preventing vascular calcification-induced calcium deposition and mitochondrial dysfunction through involvement of SIRT-1 and Nrf2.
Wang et al. [155]	Human kidney 2 cells	Until 12 h	Different doses, but no description	Pediatric acute kidney injury	RESV ↓ the inflammatory response induced by LPS in kidney cells in vitro and induced activation of Nrf2 signaling, including nuclear Nrf2 accumulation and ↑ expression of Nrf2 target genes HO-1 and NQO1.
Zhou et al. [161]	Non-tumorigenic human breast epithelial cell line MCF-10A	48 h	30 μM	Breast cancer	RESV-induced upregulation of UGT1A8 expression was abolished by silencing NRF2 gene with its specific siRNA. Likewise, RESV failed to protect DNA damage induced in Nrf2 knockdown cells, demonstrating that the protective effects for the mammary epithelial cell neoplastic transformation of RESV are based on NRF2-UGT1A8-estrogen metabolism axis.
Daverey et al. [209]	Human astrocytes-spinal cord	Until 48 h	10 mM	Protection of astrocytes against oxidative stress	RESV significantly ↑ astrocyte survival after oxidative stress, demonstrating a better effect on cell viability than curcumin and showed significant inhibition of ROS production.
Caldeira-Dias et al. [210]	Human umbilical vein endothelial cells	24 h	1 μM	Pre-eclampsia	RESV and polyphenol-rich grape juice have potentially beneficial effects on endothelial cells incubated with pre-eclampsia-derived cells, which could help manage the disease via Nrf2/ARE.
Zhou et al. [211]	Human cardiovascular endothelial cell	Until 14 h	50 mM	Endothelial dysfunction	RESV and curcumin ↑ Nrf2 translocation and Nrf2 and HO-1 protein expressions, as well as SOD activity and total cellular NAD production, than compounds alone. The results demonstrate that the combination produced a strong synergy in activity against induced oxidative stress.
Ferraresi et al. [212]	SKOV3 and OVCAR3 human ovarian cancer cells	72 h	100 μM	Ovarian cancer	RESV inhibiting the Hedgehog pathway and restoration of autophagy, counteracts LPA-induced malignancy, supporting its inclusion in the therapy of ovarian cancer for limiting metastasis and chemoresistance.
Zhang et al. [213]	Human rheumatoid arthritis fibroblast-like synoviocytes	24 h	0, 1, 10, 20, and 40 μM	Rheumatoid arthritis	RESV inhibits ROS production by activating the Nrf2-Keap1 pathway, thereby inhibiting NF-κB activation and the proliferation and migration of rheumatoid arthritis-related fibroblasts to induce apoptosis.
Chen et al. [169]	Human colon cancer LS174T and Caco2 cells	24 h	5 μM	Colitis and colon cancer	The inductor agent upregulated a wide range of endogenous Nrf2 target genes in vitro with greater potency than its parent compound RESV.
Leong et al. [214]	Cardiomyoblast cell line H9c2	96 h	30–120 microM	Prevention of cardiovascular diseases	RESV promoted entry into cell cycle arrest but extended the myogenic differentiation progress, also in modulating cell cycle control and differentiation in cardiomyoblasts.
Lu et al. [201]	Myoblast cell line	24 h	20 µM	Doxorubicin-induced cardiotoxicity (chemotherapy)	Co-treatment of RESV and FGF1 exhibits a more powerful cardio-antioxidative capacity in a model treated with doxorubicin. SIRT1/NRF2 inhibition abolished RESV in combination with FGF1 in cardioprotective action.
Sabzevary-Ghahfarokhi et al. [215]	Epithelial cells isolated from colon tissue	24 and 48 h	190 nM or 75 nM	Ulcerative colitis	Nrf2 is prevalent in inflamed tissues of patients with ulcerative colitis. RESV can reverse the inflammatory effects of TNF-α by ↓ IL-1β and ↑ IL-11 production, but Nrf2 gene expression was not altered.
Krajka-Kuźniak et al. [179]	Human keratinocites	48 h	20 or 60 μM	Human keratinocytes	RESV and its methylthioderivatives activate Nrf2 in human keratinocytes with upregulation of GST isoenzymes.
Chiang et al. [216]	Neuroblastoma cells	48 h	10 µM	Glucose oxygen deprivation	RESV rescued oxidative stress, Nrf2, and ↓ Nrf2 antioxidant target genes, like SOD, Gpx GSH, CAT, and HO-1. These protective effects of RESV are affected by blocking AMPK antagonists.
Tan et al. [217]	Epithelial-like morphology isolated from hepatocellular carcinoma cells	28 h	50 μM	Mitochondrial dysfunction and circadian cycle	Beneficial effects of RESV on stimulation of the Nrf2/NQO-1 pathway and expression of the mitochondrial respiratory complex in cells. Likewise, the inhibitory effects of RESV on NF-κB inflammation signaling were acrylamide-dependent.
Zhou et al. [180]	Epithelial cell line isolated from the mammary glands of a mouse	52 h	50 mg/kg	*Streptococcus uberis* infection	RESV can function as an activator of the p62–Keap1/Nrf2 signaling pathway to improve oxidative injury caused by S. uberis in mammary glands as well as in EpH4-Ev cells.
Zhou et al. [218]	Mammary epithelial cells	24 h	43.81 μM	Oxidative stress and apoptosis in cells contaminated by aflatoxins	RESV exhibits a good regulatory effect about expression of mRNA transcripts of Nrf2, Keap1, NQO1, HO-1, SOD2 and HSP70 for components of the Nrf2 signaling pathway.
Li et al. [219]	Human promyelocytic leukemia cells	Until 72 h	25, 50, 100, and 200 mol/L	Acute myelogenous leukemia	RESV reverses drug resistance of studied cells by regulating the PI3K/Akt/Nrf2 signaling pathway.
Zhang et al. [220]	Pheochromocytoma of the rat adrenal medulla cells	24 h	25, 50 or 100 μM	Neurotoxicity	The neuroprotective effect of the RESV analogue was achieved by several pathways, including direct elimination of ROS, rescue of endogenous antioxidants and activation of Nrf2 way antioxidant response elements.
Li et al. [221]	Breast cancer cell	24 h	RESV dimers dissolved in DMSO to give a final 0.1% solution	Elimination of ROS and Nrf2 activation	RESV and its dimers play an important role in activating the Nrf2/ARE signaling pathway.
Kima et al. [222]	Human lung cancer cells	Until 24 h	50 µM	Lung cancer	RESV-loaded nanoparticles restored H_2_O_2_-induced ROS levels by inducing cellular uptake of RESV in cells. Furthermore, RESV activated Nrf2-Keap1, thereby accumulating abundance of Nrf2.
Csiszár et al. [223]	Cerebromicrovascular endothelial cells	24 h	10 µmol/L	Antiaging effect	The treatment of aged cerebromicrovascular endothelial cells with Nrf2 activated by RESV significantly ↓ cellular production of ROS and inhibited apoptosis.
Wu et al. [191]	Neuroblastoma cells	27 h	3 µM	Cerebellar neurotoxicity in spinocerebellar ataxia type 3	RESV and caffeic acid ↑ the levels of antioxidant and autophagy protein expression with consequently corrected levels of ROS, mitochondrial membrane potential, mutant ataxin-3 in cells. Further, RESV enhanced the transcriptional activity of Nrf2.
Bigagli et al. [224]	Human granulocytes and monocytes	Until 48 h	5–10 μM	Lipopolysaccharide (LPS)-mediated cellular inflammation	An anti-inflammatory effect of RESV and hydroxy-tyrosol at nutritionally relevant low concentrations was seen, involving inhibition of granulocyte and monocyte activation and Nrf2 activation.
Moghadan et al. [225]	Human hepatocellular carcinoma cells	72 h	0, 10, 20, and 40 μM	Hepatocellular carcinoma	Low concentration of berry-derived polyphenols (RESV, gallic acid, and kuromanin chloride) upregulates hTERT expression in the hepatocellular carcinoma cell line through induction of the signaling pathway SIRT1/Nrf2.
Xuan et al. [226]	Mouse preosteoblast cell line	14 days	5 μM	Diabetic osteoporosis	Osteoblastic dysfunction under high glucose condition was significantly improved by RESV through activation of Nrf2 to suppress oxidative stress by the AKT/glycogen synthase kinase 3β (GSK3β)/FYN axis.
Li et al. [193]	Rat tubular epithelial cell	4 h	100 μmol/mL	Renal ischemia–reperfusion injury (IRI)	RESV shows renoprotection exerting significant effects on inflammatory responses, oxidative stress and apoptosis through the Nrf2/TLR4/NF-κB pathway.
Yang et al. [227]	Rats cortical neurons	24 h	10, 20, 40,60 and 80 mM	Ischemia/Reperfusion	After treatment with RESV, Nrf-2 was significantly translocated to the nuclei, expressions of Nrf-2 in the nuclei and NQO-1 and HO-1 in the cytoplasm were significantly ↑, suggesting that RESV ↑ activation of the Nrf-2/ARE signaling pathway during this injury.
Zhang et al. [228]	Pheochromocytoma of the rat adrenal medulla cells	24 h	25–75 μM	Exposure to paraquat	RESV showed neuroprotection as it reduces the response to oxidative stress and apoptosis and promotes the activity of the Nrf2 signaling pathway, cellular activity and survival rate in cells treated with paraquat.
Singh et al. [199]	Non-tumorigenic human breast epithelial cell	Until 48 h	50 µM	Breast cancer	RESV alone or in combination with 17β-estradiol significantly ↑ the expression of Nrf2 in breast tissues. Expression of antioxidant genes regulated by NRF2, NQO1, SOD3 and OGG1 that are involved in protection against oxidative DNA damage were ↑ in breast tissues treated with RESV and RESV + 17β-estradiol.
Xu et al. [200]	Primary mouse hepatocytes	24 h	20 μM	Acute liver toxicity	RESV/FGF1 ↓ oxidative stress and thereby alleviated liver injury by promoting nuclear translocation of Nrf2 and subsequently ↑ expression of antioxidant proteins in an AMPK-dependent.
Gurusinghe et al. [229]	Placental explants	Until 48 h	50, 100 or 200 mM	Trofloblastic and endothelial dysfunction	RESV, can mitigate the ↑ in placental production of sFlt-1 and activin A that occurs in response to placental injury in vitro and improve induced endothelial dysfunction in pre-eclampsia in vitro, mediated by Nrf2.

↑: increase; ↓: decrease; 5-Aza: 5-Azacytidine; AKT: serine/threonine kinase; AMPK: adenine monophosphate-activated protein kinase; ARE: antioxidant response element; C33: imazosulfuron, pesticide/herbicide; DM: diabetes mellitus; DNA: deoxyribonucleic acid; FYN: FYN Proto-Oncogene, Src Family Tyrosine Kinase; GSK3β: glycogen synthase kinase 3β; HMOX1: human gene encoding the enzyme heme oxygenase; HO-1: heme oxigenase-1; IRA-2: methoxyl-3,6-dihydroxyl; LPS: lipopolysaccharide; NF-κB: nuclear factor kappa b; Nrf2: nuclear erythroid 2-related factor 2; PI3K/Akt: phosphatidylinositol 3-kinase/serine/threonine kinase; RESV: resveratrol; ROS: oxygen reactive species; sFlt-1: soluble tyrosine kinase 1; SIRT1: sirtuin 1; STAT-3: signal transducer and activator of transcription 3; tBHQ: *tert*-butyl hydroquinone; TLR4: Toll-like receiver type 4.

**Table 3 pharmaceuticals-17-00232-t003:** List of resveratrol (RESV) uses and Nrf2 pathway-based activity in randomized clinical trials.

Author and Year	Population	Intervention and Control Groups and Sample *n*	Route of Administration	Intervention Time	Dose	Condition/Disease	Mechanism in Nrf2
Rabbani et al. [230]	Overweight and obese subjects	32 overweight and obese subjects with normal, impaired fasting, or impaired postprandial glucose, in crossover groups	Oral	8 weeks, with 6 weeks washout between crossover treatment periods	RESV 90 mg and 120 mg hesperetin	Insulin resistance in overweight and obese individuals	RESV and hesperetin showed a negative correlation between methylglyoxal levels and also peripheral blood mononuclear cell activity with quinone reductase enzyme activity, an important Nrf2 activation marker. In addition, an improvement in the inflammatory profile and insulin resistance was seen in individuals in the intervention group.
Saldanha et al. [231]	Chronic kidney disease individuals	20 nondialyzed chronic kidney disease individuals, in crossover groups	Oral	4 weeks, with 8 weeks washout between crossover treatment periods	RESV 500 mg	Chronic kidney disease	RESV supplementation in patients with chronic kidney disease not on dialysis showed no antioxidant and anti-inflammatory effect.

Nrf2: nuclear erythroid 2-related factor 2; RESV: resveratrol.

**Table 4 pharmaceuticals-17-00232-t004:** List of pterostilbene (PTS) uses and Nrf2 pathway-based activity in vitro studies.

Author and Year	Cell Culture	Intervention Time	Concentration	Condition/Disease	Mechanism Based on Nrf2
Bhakkiyalakshmi et al. [241]	HEK293T cells	24 h	I_1_: 5 µM I_2_: 10 µM	Oxidative stress	↑ dose of PTS induced a ↑Nrf2 protein. PTS activated Nrf2 expression, via direct inhibition of Keap1–Nrf2 PPI, ↓ ROS and ↑ antioxidant enzymes.
Zhou et al. [242]	HaCaT cells	24 h	I_1_: 3.75 µM I_2_: 7.5 µM I_3_: 15 µM I_4_: 30 µM	Protective effect on cytotoxicity	↓ ROS and MDA, and improved arsenic-induced SOD depletion via NRF2.
Bhakkiyalakshmi et al. [243]	INS-1E (pancreatic β-cells)	48 h	I_1_: 2 µM I_2_: 4 µM I_3_: 8 µM I_4_: 16 µM	Diabetes mellitus	Activated the Nrf2 pathway dose and time dependently, and also ↑ the expression of HO-1, SOD, CAT and GPx.
Deng et al. [244]	HUVECs	24 h	I_1_: 5 µM I_2_: 10 µM I_3_: 50 µM	UVA/UVB radiation	↑ Nrf2 levels in the nucleus of cells, and ↓ levels of carbonyl proteins and MDA.
Li et al. [245]	HaCaT cells	24 h	I_1_: 5µ M I_2_: 10µ M	Damage from UVB radiation	Attenuated UVB-induced cell death, ↓ROS, and ↑Nrf2. Aided in the repair of damaged DNA through a PI3K-dependent activation of the Nrf2/ARE pathway.
Tang et al. [246]	HUVECs	24 h	I_1_: 12.5 µM I_2_: 25 µM I_3_: 50 µM	Methylglyoxal (MGO)-induced cytotoxicity	↑ GLO-1 and GSH, suppressing oxidative stress and ↑ Nrf2, SOD, HO-1 and CAT.
Yang et al. [247]	Neuroblastoma cells SH-SY5Y	24 h	I_1_: 2.5 µM I_2_: 5 µM I_3_: 10 µM	Diabetic encephalopathy	↓ glucose, ROS and LDH levels; ↑cell viability, Nrf2, HO-1 and GST.
Lin et al. [248]	Cardiomyocyte cell line (SCC065)	24 h	I_1_: 2.5 µM I_2_: 5 µM	Cardiomyocytes	↓ PCSK9/SREBP2 interaction and mRNA expression by ↑ hsa-miR-335, hsa-miR-6825 expression, and LDLR mRNA expression.

I: intervention; ↑: increased; ↓: decreased; PTS: pterostilbene; PPI: protein–protein interaction; Nrf2: nuclear factor erythroid 2-related factor; ROS: reactive oxygen species; Keap1: Kelch sample-related protein1; MDA: malondialdehyde; SOD: superoxide dismutase; HO-1: heme oxygenase-1; CAT: catalase; GPx: glutathione peroxidase; HUVECs: human umbilical vein endothelial cells; GLO-1: glyoxalase-1; HaCaT: cell line of aneuploid immortal keratinocytes from adult human skin; GSH: glutathione; GST: glutathione-S-transferase.

**Table 5 pharmaceuticals-17-00232-t005:** List of pterostilbene (PTS) uses and Nrf2 pathway-based activity in animal models.

Author and Year	Sample (*n*)	Route of Administration	Intervention	Condition	Mechanism in Nrf2
Xu et al. [249]	8 BALB/c mice/group	Itraperitoneal injection	C: OVA 10 μg + 200 μL of saline solution I_1_: OVA + 30 mg/kg PTS I_2_: OVA + 50 mg/kg PTS	Asthma	↓ inflammation and IL-4, IL-13, IL-5, MDA, while ↑ SOD and CAT, regulating the AMPK/Sirt1 and Nrf2/HO-1 signaling pathway
Xu et al. [250]	10 Swiss–Kunming mice/group	Intracerebroventricular injection (ICV)	I_1_: ICV Aβ1–42 + 10 mg/kg PTS I_2_: ICV Aβ1–42 + 20 mg/kg PTS I_3_: ICV Aβ1–42 + 40 mg/kg PTS	Cognitive dysfunction	↓ loss of neurons and ROS in Aβ1–42; ↑ SOD and HO-1; and promotes Keap1-associated protein-1 and p62 binding that, ↑Nrf2 activation.
Kosuru et al. [251]	8 Sprague Dawley rats/group	Oral	C: 65% corn starch I_1_: 20 mg/kg/day of PTS I_2_: 20 mg/kg/day of PTS + 65% high-fructose diet	Cardiovascular and inflammatory diseases associated with diabetes	↓cardiac hypertrophy, hypertension, oxidative stress, inflammation, NF-κb expression and NLRP3 inflammasome; and ↑Nrf2, HO-1, via AMPK/Nrf2/HO-1.
Bhakkiyalakshmi et al. [241]	6 swiss albino mice/group	Intraperitoneall	C: Healthy I_1_: Healthy + PTS (10 mg/kg) I_2_: Diabetic (STZ 50 mg/kg) I_3_: Diabetic + PTS (5 mg/kg) I_4_: Diabetic + PTS (10 mg/kg) I_5_: Diabetic + GB (600 µg/kg)	Diabetes	Regulated glycemia, insulin, and lipoproteins (VLDL-c, LDL-c, HDL-c). Furthermore, ↓ LPO, and ↑ Nrf2, SOD, CAT, GSH, gpx, via Keap1/Nrf2.
Hseu et al. [252]	Zebrafish embryos	Immersion	C: DMSO (0.1%) I_1_: PTS 2.5 µM I_2_: PTS 5.0 µM	Melanogenesis	↓ ROS, activating the Nrf2, leading to ↑ HO-1, γ-GCLC and NQO-1. Also, inhibited tyrosinase expression/activity and endogenous pigmentation in the zebrafish model.
Xue et al. [253]	6 Sprague Dawley rats/group	P.O.	C_1_: ACL + sunflower oil C_2_: ACLX + sunflower oil I_1_: ACLX + PTS (30 mg/kg)	Osteoarthritis	↓COX-2, INOs, PGE2, NO, and ROS, also ↑Nrf2, inhibits IL-1β-induced inflammation in chondrocytes.
Zhang et al. [254]	10 BALB/c mice/group	Intraperitoneal injection	C_1_: PBS I_1_: PTS (50 mg/kg) I_2_: LPS (30 µg/kg) + *D*-GalN (600 mg/kg) I_3_: PTS (12.5 mg/kg) + L/D I_4_: PTS (25 mg/kg) + L/D I_5_: PTS (50 mg/kg) + L/D	Acute liver failure	Protected against acute liver failure, ↓ lethality, and ALT, AST, IL-6, IL-1b, TNF-α, MDA and MPO, improved histology from the liver, and ↑ GSH and SOD. Furthermore, ↓ NLRP3, MAPK and NF-κb pathways, and ↑ expression of the NRF2 pathway.
Zhang et al. [255]	10 BALB/c mice/group	Intraperitoneal injection	C: Healthy I_1_: LPS-induced I_2_: LPS + PTS (10 mg/kg) I_3_: LPS + PTS (20 mg/kg) I_4_: LPS + PTS (40 mg/kg) I_5_: LPS + DEX (5 mg/kg)	Acute lung injury	Improve acute lung injury, ↓ MPO, COX-2, inos, TNF-α, IL-6, IL-1β and MDA induced by LPS and ↑ SOD, CAT and GSH-Px, via Nrf2/HO-1/ARE.
Benlloch et al. [256]	nu/nu nude mice	Intravenous administration	I_1_: *A2058* + PTS (50 mg/mL) I_2_: *MeWo* + PTS (50 mg/mL) I_3_: *MelJuso* + PTS (50 mg/mL)	Cancer	↓ the growth of human melanoma, pancreatic cancer and ROS in vivo. Also ↑ expression of the NRF2 pathway.
Fan et al. [257]	5 C57BL/6 mice/group	Intraperitoneal injection	C: saline solution I_1_: PTS (50 mg/kg I_2_: APAP (900 mg/kg) I_3_: APAP (900 mg/kg) + DMSO (0.05 mL/kg) I_4_: APAP (900 mg/kg) + PTS (50 mg/kg)	Hepatotoxicity	↓ ALT, AST, histological lesions, MDA, and MPO. Also, ↑ liver GSH and SOD levels, and was able to activate the Nrf2/HO-1 signaling pathway.
Lacerda et al. [258]	7–8 *Wistar* adult rats/group	-	C: IAM I: IAM PTS + PBS (100 mg/kg)	Myocardial infarction	PTS + PBS, ↓ LPO and trxr, and ↑ GST and grx. Additionally, ↑ the expression of Nrf2 and p-GSK-3b, while ↓ p-GSK-3b/GSK-3b ratio of infarcted animals, promoting an improvement in systolic function post-infarction.
Zeng et al. [259]	14 Rat pups *Sprague Dawley*/ group	Oral gavage	C: Healthy I_1_: 12.5 mg/kg I_2_: 25 mg/kg I_3_: 50 mg/kg	Hypoxic–ischemic brain injury	↓ NF-κb, NO, TNF-α, IL-1β, IL-6, ROS and MDA, furthermore, ↑ GSH and regulated the NRF2/HO-1 and JNK pathway and activated PI3K/Akt-mTOR signals.
Liu et al. [260]	BALB/c mice	Injected intraperitoneally	C: Healthy I_1_: LPS (50μg/kg) + D-Gal (500 mg/kg) I_2_: PTS (10 mg/kg) + LPS (50 μg/kg) + D-Gal (500 mg/kg) I_3_: PTS (20 mg/kg) + LPS (50 μg/kg) + D-Gal (500 mg/kg) I_4_: PTS (40 mg/kg) + LPS (50 μg/kg) + D-Gal (500 mg/kg) I_5_: PTS (40 mg/kg)	Acute liver injury	↓ LPS/D-Gal-induced inflammatory infiltration, hemorrhage and hepatic cord dissociation by reducing MPO activity in the liver. In addition ↓ALT, AST, TNF-α, IL-6, IL-1β and NF-κb, also upregulated Nrf2 and HO-1.
Millán et al. [261]	12 New Zealand rabbits/group	Subcutaneously administered	C: non-diabetic I_1_: diabetic (alloxan—100 mg/kg) I_2_: diabetic + PTS (50 mg/kg)	Retinopathy caused by diabetes	↓ ROS, and rate of cell death, and activate the PI3K/AKT/GSK3β/NRF2 pathway, ↑ SOD, CAT, GPX, and GSH/GSSG ratio.
Obrador et al. [262]	Wild-type B6SJLF1/J Mice	Orally	I_1_: Nicotinamide riboside (185 mg/kg) I_2_: PTS (30 mg/kg)	Amyotrophic lateral sclerosis	↑ Survival and improved loss of neuromotor functions associated with ALS in transgenic mice
Dornadula et al. [263]	Swiss albino mice	Intraperitoneally administered	C: Healthy I_1_: Healthy + PTS (5 mg/kg) I_2_: Diabetic (STZ—50 mg/kg) I_3_: Diabetic + PTS (5 mg/kg) I_4_: Diabetic + glibenclamide (600 μg/kg)	Diabetes	↑ Nrf2, SOD, CAT, GPx, NQO-1 and HO-1, ↓ NO production and inhibition of INOS activation.
Tang et al. [264]	12 Sprague Dawley rats/group	Orally	C: PBS I: PTS (10 mg/kg)	Atherosclerosis	↓ Inflammatory response, atherogenesis, aortic plaque size, macrophage infiltration, oxidative stress and apoptosis cell. Also, ↑ Nrf2, and ↓ stat3.
Xiong et al. [265]	6–10 Sprague Dawley rats/group	Orally	C: Healthy I_1_: phosfate buffer saline (PBS) I_2_: PTS (10 mg/kg)	Atherosclerosis	↓ HDL-c, TC, LDL-c, IL-1, TNF-α and IL-6. Furthermore, ↑ endothelial cell viability and regulated apoptosis by activating the Nrf2 pathway via the TLR-4/myd88/NF-κb pathway.
Yang et al. [266]	5 BALB/C mice/group	Injected intraperitoneall	C: 1% DMSO I_1_: LPS (1 mg/kg) I_2_: PTS (50 mg/kg) I_3_: LPS + PTS (12.5 mg/kg) I_4_: LPS + PTS (25 mg/kg) I_5_: LPS + PTS (50 mg/kg)	Pulmonary fibrosis	↓ NF-κb, NOD-type NLRP3 receptor, MPO, TNF-α, IL-6, MDA, and IL-1β, and ↑ IL-10, SOD, and GSH. Also, activated Nrf2 nuclear translocation and activated HO-1 and NAD(P)H genes.
Yao et al. [267]	10 Kunming male mice/group	Intragastric administration	C: CMC-Na (10 mL/kg) STC: LOP (5 mg/kg) + CMC-Na (10 mL/kg) I_1_: LOP (5 mg/kg) + PTS (30 mg/kg) I_2_: LOP (5 mg/kg) + PTS (60 mg/kg)	Laxative effect	Improved intestinal motility disorder, ↓ apoptosis, ROS, MDA, and ↑ PI3K/AKT and Nrf2/HO-1 signaling, GPx in addition, to promoting the secretion of the intestinal hormone, gastrin and motilin, and restoring microbial diversity.
Zhu et al. [268]	KM mice	Intragastric administration	I_1_: Aβ_25–35_ (9 nmol/3 μL) + PTS (10 mg/kg) I_2_: Aβ_25–35_ (9 nmol/3 μL) + PTS (40 mg/kg)	Alzheimer’s	Showed more effective neuroprotective effects against cognitive dysfunction, and improved neuronal plasticity and alleviated neuronal loss both in vivo. Also, ↑ SIRT1, Nrf2 and SOD.

I: intervention; C: control; ↑: increased; ↓: decreased; PTS: pterostilbene; IL: interleukin; Nrf2: nuclear factor erythroid 2-related factor; ROS: reactive oxygen species; MDA: malondialdehyde; SOD: superoxide dismutase; CAT: catalase; AMPK: AMP-activated protein kinase; Sirt1: silent information regulator transcript-1; HO-1: heme oxygenase-1; Keap1: Kelch sample-related protein1; OVA: ovalbumin; NLRP3: NOD-, LRR- and pyrin domain-containing protein 3; STZ: streptozotocin; GB: glibenclamide; LPO: lipid peroxidation; VLDL-c: very low-density lipoprotein-cholesterol; LDL-c: low-density lipoproteins-cholesterol; HDL-c: high-density lipoproteins-cholesterol; GPx: glutathione peroxidase; GSH: glutathione; γ-GCLC: glutamate-cysteine ligase catalytic subunit; NQO-1: quinone oxidoreductase 1; ACL: anterior cruciate ligament; ACLX: transection and partial medial meniscectomy; COX-2: cyclooxygenase-2; iNOS: induced nitric oxide synthase; PGE2: prostaglandin E2; D-GalN: D-galactosamine; L/D: lipopolysaccharide and *D*-galactosamine; DEX: dexamethasone; LPS: lipopolysaccharide; MPO: myeloperoxidase; ALT: alanine aminotransferase; AST: aspartate aminotransferase; APAP: acetaminophen; PBS: hydroxypropyl-b-cyclodextrin; TRxR: thioredoxin reductase; IAM: infarcted nontreated animals; TLR4: Toll-like receptor 4; NF-ΚB: nuclear factor kappa B; TNF-α: tumor necrosis factor-α; STC: slow-transit constipation; LOP: loperamide; CMC-Na: carboxymethylcellulose sodium; Aβ_25–35_: β-amyloid_25–35_.

**Table 6 pharmaceuticals-17-00232-t006:** List of piceatannol (PIC) uses and Nrf2 pathway-based activity in animal models.

Author and Year	Sample (*n*)	Route of Administration	Intervention Time	Dose	Condition/Disease	Mechanism in Nrf2
Wang et al. [276]	ICR mice	Intraperitoneally	24 h	I: 5, 10 or 20 mg/kg	Cerebral ischemia–reperfusion injury	↓ MDA and LDH, ↑ SOD, GSH-Px,HO-1 and NQO1
Binmahfouz et al. [277]	6 Wistar rats/group	Oral gavage	4 consecutive weeks	I: 5 or 10 mg/kg	Endometrial hyperplasia	Inhibited lipid peroxidation, ↑ SOD, CAT, and HO-1, ↓ MDA. Protected the increase in inflammatory markers ↓ IL-6, TNF-α and NF-κB
Zhang et al. [278]	6 Kunming mice/group	Orally administered	8 weeks	I: 20 mg/kg	Behavioral disorders and neurological deficits	↓ MDA levels and ↑ SOD, CAT and ↑ the expression of Nrf2, HO-1 and NOQ1
Shi et al. [279]	8 Wistar rats/group	Oral gastric gavage	28 days	I: 10 mg/kg	Testicular health, spermatogenesis and steroidogenesis	↑ NQO1, HO-1, γGCS and GPx, SOD, CAT, ↓MDA
Li et al. [280]	8 Sprague Dawley rats/group	Intraperitoneal injection	12 weeks	I: 5 or 10 mg/kg	Diabetic cardiomyopathy	↓ IL-6 and TNF-α, ↑ Nrf2 expression, via Nrf2/HO-1 and SOD
Wahdan et al. [281]	15 Wistar rats/group	Intraperitoneal injection	7 days	I: 10 mg/kg	Cisplatin nephrotoxicity	↑ HO-1, GCLC and GCLM, GSH. ↓ NF-κB, (IL-1β, TNF-α, iNOS, COX-2, MDA

↑: increased; ↓: decreased; PIC: piceatannol; Nrf2: nuclear factor erythroid 2-related factor; ROS: reactive oxygen species; LPO: lipid peroxidation; HO-1: heme oxygenase-1; NF-κB: nuclear factor kappa B; NQO1: quinone oxidoreductase 1; MDA: malondialdehyde; SOD: superoxide dismutase; GSH: phospholipid hydroperoxide glutathione; GLO-1: glyoxalase I; GPx: glutathione peroxidase; GCLC: glutamate-cysteine ligase catalytic subunit; TNF-α: tumor necrosis factor-α; IL-6: interleukin-6; iNOS: induced nitric oxide synthase; COX-2: cyclooxygenase-2; γGCS: γ-glutamyl cysteine synthetase; LDH: lactate dehydrogenase; PMs: primary peritoneal macrophages; LPS: lipopolysaccharide; AKT: protein kinase B; NO: nitric oxide; CAT: catalase; GCLM: glutamate cysteine ligase modifier subunit; DMSO: dimethyl sulfoxide; SNEDDS: self-nanoemulsifying drug delivery system; CMC: carboxymethyl celulose; ED = edaravone.

**Table 7 pharmaceuticals-17-00232-t007:** List of piceatannol (PIC) uses and Nrf2 pathway-based activity in vitro studies.

Author and Year	Cell Culture	Intervention Time	Concentration	Condition/Disease	Mechanism in Nrf2
Hao et al. [282]	Cells ARPE-19	24 h	15 μM	Prevent macular degeneration	↑ antioxidant genes, catalytic subunit glutamate-cysteine ligase (GCLc), SOD and HO-1, ↓ ROS
Kil et al. [274]	Endothelial cells	ECs were pre-incubated for 6 h with Pic before 12 h exposure to 3 mM Hcy	10 μM	Prevent endothelial cell apoptosis	Induced HO-1 expression, ↓ ROS
Achy-Brou et al. [283]	RAW 264.7 macrophages cells	24 and 48 h	3, 10, 20 or 30 μM	Cytotoxicity and ability to reduce NO	Are cytotoxic to transformed RAW 264.7 macrophages inhibit ↓ NO expression via Nrf2
Zhu et al. [284]	Human keratinocyte cells (HaCaT cell line)	24 h	15 μM	Preventing the proliferation of acne vulgaris	↓ NF-κB, ↑ HO-1 and NQO1
Li et al. [280]	H9C2 rat cardiac myoblasts	48 h	10 μM	Diabetic cardiomyopathy	↓ IL-6 and TNF-α, ↑ Nrf2 expression, via Nrf2/HO-1, SOD
Hosoda et al. [206]	C2C12 myoblasts	24 h	10 μM	Antioxidant and antiapoptotic effects	↓ ROS, ↑ HO-1
Wang et al. [285]	Highly differentiated rat adrenal pheochromocytoma cells (PC12 cells)	24 h	2.5; 10 and 40 μM	Cerebral ischemia–reperfusion injury	↓ MDA and LDH, ↑ HO-1 and NQO1

↑: increased; ↓: decreased; Ecs: endothelial cells; PIC = piceatannol Nrf2: nuclear factor erythroid 2-related factor; ROS: reactive oxygen species; HO-1: heme oxygenase-1; NF-κB: nuclear factor kappa B; NQO1: quinone oxidoreductase 1; MDA: malondialdehyde; SOD: superoxide dismutase; GCLC: glutamate-cysteine ligase catalytic subunit; TNF-α: tumor necrosis factor-α; IL-6: interleukin-6; iNOS: induced nitric oxide synthase; LDH: lactate dehydrogenase; PMs: primary peritoneal macrophages; LPS: lipopolysaccharide; HaCaT: cell line of aneuploid immortal keratinocytes from adult human skin; NO: nitric oxide.

**Table 8 pharmaceuticals-17-00232-t008:** List of pinosylvin (PIN), stilbestrol (DHS) uses and Nrf2 pathway-based activity in vitro studies.

Author and Year	Cell Culture	Intervention Time	Concentration	Condition/Disease	Mechanism in Nrf2
Koskela et al. [298]	The human retinal pigment epithelial (ARPE-19) cell line	24 h	5 μM	Cell death induced by oxidative stress	PIN, ↑ HO-1, it was observed in this work that Nrf2 levels did not show variations in the time intervals analyzed, maintaining basal levels
Wang et al. [299]	Normal human lung epithelial Beas-2B cells, human breast carcinoma MDA-MB-231 cells and RAW 264.7 cells	24 h	0.5–4 μM	Chronic obstructive pulmonary disease	DHS activates Nrf2-mediated defensive response, treatment with the compound positively regulated the levels of NQO1 and GCLM, ↑ Nrf2 and its regulated genes

↑: increased; PIN: pinosylvin; Nrf2: nuclear factor erythroid 2-related factor; HO-1: heme oxygenase-1; DHS: *Trans*-4,4′-dihydroxy stilbene; NQO1: quinone oxidoreductase 1; GCLM: glutamate cysteine ligase modifier subunit.

**Table 9 pharmaceuticals-17-00232-t009:** List of pinosylvin (PIN) and stilbestrol (DHS) uses and Nrf2 pathway-based activity in animal models.

Author and Year	Sample *(n*)	Route of Administration	Intervention Time	Dose	Condition/Disease	Mechanism in Nrf2
Wang et al. [299]	18 C57BL/6 mice/group	Intraperitoneal injection	16 weeks	I: 2 or 4 mg/kg	Chronic obstructive pulmonary disease (COPD)	DHS activates Nrf2-mediated defensive response, treatment with the compound positively regulated the levels of NQO1 and GCLM, ↑ Nrf2 and its regulated genes
Wang et al. [300]	12–13 mice/group	Intragastric administration	2 weeks	I: 100 mg/kg	Oligoasthenospermia	PIN and DHS ↓ ROS and MDA, through activation of the Nrf2/ARE pathway

↑: increased; ↓: decreased; DHS: *Trans*-4,4′-dihydroxy stilbene; PIN: pinolsilvin; REVS: resveratrol Nrf2: nuclear factor erythroid 2-related factor; ROS: reactive oxygen species; HO-1: heme oxygenase-1; NQO1: quinone oxidoreductase 1; MDA: malondialdehyde; GCLM: glutamate cysteine ligase modifier subunit; ARE: antioxidant response activity; BUS: busulfan; Dex: dexamethasone.

**Table 10 pharmaceuticals-17-00232-t010:** List of polidatin (PDT) uses and Nrf2 pathway-based activity in animal models.

Author and Year	Sample (*n*)	Route of Administration	Intervention	Condition	Mechanism in NRF2
Cao et al. [307]	30 C57BL/6 mice/ group	Intraperitoneal injection	C: non-irradiated + saline control I_1_: irradiation + saline I_2_: irradiation + PDT (100 mg/kg) I_3_: irradiation + WR2721 (100 mg/kg)	Radiation-induced lung injury	↓ Acute inflammation and fibrosis caused by radiation. Also ↑ SIRT3, Nrf2 and PGC-1α.
Zhao et al. [308]	8 Sprague-Dawley rats/group	Intragastric administration	C: control vehicle I_1_: fructose vehicle I_2_: fructose + PDT (7.5 mg/kg) I_3_: fructose + PDT (15 mg/kg) I_4_: fructose + PDT (30 mg/kg) I_5_: fructose + pioglitazone (4 mg/kg)	Fructose-induced non-alcoholic fatty liver disease	Activate the Nrf2 antioxidant pathway, suggest that augmentation of mir-200a to control the Keap1/Nrf2 pathway is a therapeutic strategy for fructose-associated liver inflammation.
Chen et al. [309]	14 C57BL/6 mice/group	Oral gavage	C: sham group I_1_: PDT I_2_: DSS (5% g/mL) I_3_: DSS + PDT	Ulcerative colitis	↓ TNF-α, IL-4, IL-6, CPX-2, inos, NF-κb, p65, and p38, while↑ IL-10, Nrf2, HO-1 and NQO1expression. It also improved intestinal inflammatory response and colonic epithelial barrier dysfunction.
Zhao et al. [310]	18 Wistar rats/group	Intraperitoneal injection	C: model group I_1_: PDT (50 mg/kg) I_2_: sham (only stripping blood vessels without ligation + saline)	Neurological function	↓ NO and MDA, and ↑ SOD, GSSG and GSH in brain tissue. Furthermore, ↑ Nrf2, NQO1 and HO-1, and ↓ oxidative stress, through the NRF2/ARE pathway.
Lv et al. [311]	8 Sprague Dawley rats/group	Intraperitoneal injection	C: Sham I_1_: SCI I_2_: SCI + PDT 20 mg/kg I_3_: SCI + PDT 40 mg/kg	SCI	↓ ROS, LDH and apoptosis status via NRF2/HO-1 in addition to regulating the expression of Nrf2.
Huang et al. [306]	12 Wistar rats/group	Intragastric administration	C: Healthy I_1_: ethanol I_2_: silymarin (100 mg/kg) I_3_: PDT (25 mg/kg) I_4_: PDT (50 mg/kg) I_5_: PDT (100 mg/kg)	Parkinson’s disease	↑ p-AKT, p-GSK-3βser9 and Nrf2 levels, and suppressed NF-κb activation, via AKT/GSK3β-NRF2.
Li et al. [312]	12 BALB/c mice/ group	Intraperitoneal injection + vaginal	C: Healthy I_1_: LPS I_2_: LPS + PDT (20 mg/kg) I_3_: LPS + PDT (40 mg/kg) I_4_: LPS + PDT (80 mg/kg)	Endometriosis	↓TNF-α, IL-1β, and IL-6, NF-κb activation in a dose-dependent manner. Also, ↑ Nrf2 and HO-1.
Zhang et al. [313]	8 ICR mice/ group	Intragastrically administered	C: without treatment I_1;_ SM (40 mg/kg) I_2_: SM + PDT (100 mg/kg) I_3_: SM + PDT (200 mg/kg) I_4_: SM + PDT (400 mg/kg) I_5_: NAC (200 mg/kg)	Sulfur mustard-induced liver injury	↑ Sirt1, HO-1, NQO1 and Nrf2, via the Sirt1/Nrf2 pathway.
Bheereddy et al. [314]	Sprague Dawley rats	Orally	I_1_: PDT (25 mg/kg) I_2_: PDT (50 mg/kg))	Diabetes mellitus	↑ SIRT1 driven by PGC-1α, facilitating Nrf2-directed antioxidant signaling.
Gu et al. [315]	C57BL/6 mice	Intraperitoneal injection	C: Healthy I_1_: PDT (20 mg/kg) I_2_: PDT (40 mg/kg) I_3_: PDT (80 mg/kg) I_4_: LPS (10 mg/kg)	Acute kidney injury	↓ Creatinine, TNF-α, IL-1β, IL-6, MDA, and NF-κb, increased Nrf2 and HO-1 expression.
Li et al. [316]	8 BALB/c mice/group	Intragastric	C: Healthy I_1_: Graves’ orbitopathy I_2_: Graves’ orbitopathy + PDT (50 mg/kg) I_3_: PDT (50 mg/kg)	Ocular injury	Attenuated orbital muscle adipose tissue expansion and lipid droplet accumulation through the Keap1/Nrf2/ARE pathway. It also ↓ ROS induced by H_2_O_2_, and ↑ the expression of NAD(P)H and NQO1.
Gu et al. [317]	8 Sprague Dawley rats/group	-	C: standard diet + water I_1_: fructose-vehicle group receiving PBS I_2_: PDT (7.5 mg/kg) I_3_: PDT (15 mg/kg) I_4_: PDT (30 mg/kg) I_5_: chloroquine (50 mg/kg)	Fructose-induced kidney injury	↑ Nrf2, antioxidant activity, and suppressed autophagy to protect against fructose-induced podocyte injury.
Zhan et al. [318]	C57BL/6J mice	-	I_1_: PDT (15 µMol/L) I_2_: PDT (30 µMol/L)	Ischemia/reperfusion injury of the spinal cord	↑ Neuronal viability and protected against apoptosis and mitochondrial injury in a dose-dependent manner. Mechanistically, downregulated keap1 and ↑ nrf2, nqo-1 and ho-1, and also reversed neuronal and mitochondrial damage in a mouse model.
Chen et al. [319]	Sprague Dawley rats	Intraperitoneal injection	I_1_: PDT (10 mg/kg) I_2_: PDT (100 mg/kg) I_3_: PDT (250 mg/kg) I_4_: PDT (500 mg/kg)	Acute Myocardial Infarction (AMI)	↓ hypoxia-induced H9c2 cell apoptosis and ROS generation, by ↑ Nrf2/HO-1 signaling.
Chen et al. [295]	8 Sprague Dawley rats/group	Intraperitoneal injection	C: non-PDT treatment (1% Tween 80) I_1_: PDT (25 mg/kg) I_2_: PDT (50 mg/kg)	Sciatic nerve in diabetes mellitus	↓ ROS and upregulated the expression of Nrf2 and GLO1 and inhibited the expression of Keap1 and RAGE.
Gong et al. [320]	12 C57BL/6J mice/group	Intraperitoneal injection	C: diabetic model (STZ 50 mg/kg) I_1_: Diabetic + PDT (100 mg/kg) I_2_: Diabetic + PDT (200 mg/kg) I_3_: Diabetic + metformin (195 mg/kg)	Diabetes mellitus	↑ activity of Nrf2, CKIP-1, and antioxidant effect. Suggested that PDT could ↑ the CKIP-1-Nrf2-ARE pathway to prevent OSS-induced insult in gmcs and diabetic mice.
Huang et al. [321]	12 Wistar rats/group	Intragastrically	C: non-PDT treatment I_1_: ethanol I_2_: silymarin (100 mg/kg) I_3_: PDT (25 mg/kg) I_4_: PDT (50 mg/kg) I_5_: PDT (100 mg/kg)	Alcoholic liver injury	↑ Nrf2 and HO-1. In addition, ↓TNF-α, IL-1 β and IL-6 through downregulation of TLR4 and NF-κB.
Tong et al. [322]	6 Sprague Dawley rats/group	Intraperitoneal injected	C: Sham operated I_1_: doxorubicin-treated (2 mg/kg) I_2_: PDT (50 mg/kg) I_3_: doxorubicin (2 mg/kg) + PDT (50 mg/kg)	Cognitive impairment induced by chemotherapy (chemobrain)	Protected against learning and memory impairment by restoring the histopathological architecture of the hippocampus. Furthermore, it suppressed oxidative stress through the upregulation of nrf2.
Tang et al. [323]	5 C57BL/6 wild-mices/ Group	Intraperitoneal injection	C: Sham operated I_1_: surgical destabilization of the medial meniscus I_2_: surgical destabilization of the medial meniscos + PDT (100 mg/kg)	Osteoartitis	Suppressed the overproduction of pro-inflammatory mediators, including PGE2, TNF-α, NO, COX-2, inos and IL-6 in IL-1β-induced chondrocytes.
Li et al. [324]	7 guinea pigs/group	Intraperitoneal injection	C: non-PDT treatment I_1_: cisplatin (12 mg/kg) I_2_: 2% DMSO + cisplatin (12 mg/kg) I_3_: PDT (40 mg/kg) I_4_: dexamethasone (10 mg/kg) + cisplatin (12 mg/kg) I_5_: PDT (20 mg/kg) + cisplatin (12 mg/kg) I_6_: PDT (40 mg/kg) + cisplatin (12 mg/kg) I_7_: PDT (80 mg/kg) + cisplatin (12 mg/kg)	Hearing loss	↓ ABR, promote Nrf2 nuclear translocation, ↑ Nrf2 and HO-1 expression, and, thus reducing the loss of ohcs.
Zeng et al. [325]	BALB/c mice	Intraperitoneal injection	C: saline solution I_1_: ovalbumin (20 µg) + PDT (100 mg/kg)	Asthma	↓ lung ROS, TGF-β1 expression and fibroblasts. In addition to reversing the expression of (NOX), promoting the expression of HO-1 mediated by Nrf2.
Huang et al. [326]	10 Sprague Dawley rats/group	Gavage	C: healthy I_1_: PDT (150 mg/kg)	Diabetic nephropathy	↓ the expression of FN and TGF-β1 exposed to ages. It also ↑ Nrf2, HO-1 and SOD1. Activation of the Nrf2-ARE pathway by PDT led to suppression of the overproduction of ROS markedly driven by ages.
Ye et al. [327]	3–5 BALB/c mice/ Group	-	C: healthy I: PDT (40 µM)	Allergy	↓ TNF-α, IL-4, IL-1β and IL-8, and downregulated the downstream signaling pathway including MAPK, PI3K/AKT and NF-κb. It also targets the Nrf2/HO-1 pathway to inhibit mast cell-derived allergic inflammatory reactions.

I: intervention; C: control; ↑: increased; ↓: decreased; PDT: polydatin; IL: interleukin; Nrf2: nuclear factor erythroid 2-related factor; ROS: reactive oxygen species; MDA: malondialdehyde; SOD: superoxide dismutase; CAT: catalase; PGC-1α: proliferator-activated receptor gamma-coactivator-1α; Sirt3: silent information regulator transcript-3;LPO: lipid peroxidation; HO-1: heme oxygenase-1; NF-ΚB: nuclear factor kappa B; NQO1: quinone oxidoreductase 1; iNOS: induced nitric oxide synthase; COX-2: cyclooxygenase-2; Keap1: Kelch sample-related protein1; TLR4: Toll-like receptor 4; MPO: myeloperoxidase; MDA: malondialdehyde; SOD: superoxide dismutase; GSH: phospholipid hydroperoxide glutathione; GLO-1: glyoxalase I; Sirt1: silent information regulator transcript-1; GPx: glutathione peroxidase; PI3K: phosphoinositide 3-kinase; TNF-α: tumor necrosis factor-α; IL-6: interleukin-6; LDH: lactate dehydrogenase; TG: triglycerides; PGE2: prostaglandin E2; AKT: protein kinase B; GSK-3β: activated glycogen synthase kinase-3 beta; AST: aspartate aminotransferase; CK: creatine kinase; NO: nitric oxide; DSS: dextran sodium sulfate; SCI: spinal cord injury; SM: sulfur mustard; NAC: N-acetyl-L-cysteine.

## Data Availability

Data sharing is not applicable.

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
