# Peer review of "E-Stilbenes: General Chemical and Biological Aspects, Potential Pharmacological Activity Based on the Nrf2 Pathway"

_pharmaceuticals, 2024, doi:10.3390/ph17020232_

Round 1
Reviewer 1 Report
Comments and Suggestions for Authors
The present manuscript by Elaine L. S. S. Mendonça et al is a comprehensive review of stilbenes: chemistry, biosynthesis, metabolism, pharmacological activities and their role in activating the Nrf2 pathway. Overall this review could be of interest to researchers in the field. However, it is very difficult to understand and should be carefully edited and possibly shortened by including/discussing only the most important findings/references.
Below I mention five examples of specific comments which could help the authors in revising their manuscript:
1) Line 22, “…The last route involved metabolic engineering to enhance production. It is thought that these activities can be modulated in the human body…”. Which activities?
2) Line 730, “…Among the studies listed in the table 6, Achy-Brou et al. [204] evaluated…”. In table 6 there is no Achy-Brou et al. [204].
3) Line 778, “Wang et al. [297] evaluated the activation of the Nrf2 pathway in…”. There is no Wang et al. [297] in either Table 8 or 9.
4) Line 779, “…The concentration tested in this study was 100 mg/kg…” should change to“…The dose tested in this study was 100 mg/kg…”.
5) In all tables presented in vitro results, in the column referring as Dose should change to Concentration.
Extensive editing of English language required.
Author Response
REVIEWER #1:
Thanks. Very good remarks and careful analysis. Sorry for our mistakes.
- “Below I mention five examples of specific comments which could help the authors in revising their manuscript”:
- Line 22, “…The last route involved metabolic engineering to enhance production. It is thought that these activities can be modulated in the human body…”. Which activities?
Answer: Line 22 is in the summary, as it is a section that brings together the main information of the manuscript in a succinct and objective way. However, this specific issue is covered in depth in the topic "2.1 Stilbenes’ biosynthesis" (line 130). In the summary, we are confirming that the biosynthesis of stilbenes can currently follow two routes, the first and main one, being the natural route, synthesized by plants, and the second, synthesized by microorganisms, this being an alternative route to quantitatively optimize the synthesis of stilbenes. Still in the summary session, when we stated that "It is believed that in the human body, these potential activities can be modulated by the regulation of the nuclear factor erythroid-derived 2 (Nrf2), which increases the expression of antioxidant enzymes.", we did not go into depth, because we brought 4 topics (2.1 Stilbenes' biological metabolism - line 175; 3. Stilbenes: diseases-based biological and pharmacological activities - line 266; 4. Stilbenes' role in activating the Nrf2 pathway - line 429; and 5. Stilbenes: compounds-based approach - Line 557) that develop these activities and mechanisms in detail.
- Line 730, “…Among the studies listed in the table 6, Achy-Brou et al. [204] evaluated…”. In table 6 there is no Achy-Brou et al. [204].
Answer: In fact, the study by Achy-Brou et al. [283] was not in table 6, but in table 7. We have already corrected this mistake in line 731.
"Among the studies listed in Table 7, Achy-Brou et al. [283] evaluated the activation of Nrf2 through the reduction in ●NO, differing from other researchers who generally evaluated the production of antioxidant enzymes related to the activation of the Nrf2 pathway.”
Line 778, “Wang et al. [297] evaluated the activation of the Nrf2 pathway in…”. There is no Wang et al. [297] in either Table 8 or 9.
Answer: In fact, the study cited in the text was that of Wang et al. [297], but the one in the table was Wang et al. [300], we believe there was a problem with the referral system. But we have already manually fixed this issue. The study by Wang et al. [300] is mentioned in table 9. We fixed it in the text, on line 778.
“Wang et al. [300] evaluated the activation of the Nrf2 pathway in oligoasthenospermia for three stilbenes: PIN, DHS, and REVS.”
Line 779, “…The concentration tested in this study was 100 mg/kg…” should change to”…The dose tested in this study was 100 mg/kg…”
Answer: We have corrected them in the text, line 779.
"The dose tested in this study was 100 mg/kg, a much higher level compared to other studies carried out."
- In all tables presented in vitro results, in the column referring as Dose should change to Concentration.
Answer: We replaced them in all tables referring to in vitro studies.
Extensive editing of English language required.
Answer: The text was fully reviewed by MDPI services. We added the certificate.
Reviewer 2 Report
Comments and Suggestions for Authors
The article seems well structured. There are no major errors to report. It is suggested to continue the research by highlighting how the stereochemistry of the double bond can influence the reactivity of stilbene at the cellular level. The stereochemistry of the entire system could be different if evaluated not only from the point of view of oxidative stress but also as an interaction in enzymatic systems.
Author Response
REVIEWER #2:
Thanks. Very good remarks and careful analysis. Sorry for our mistakes.
- The article seems well structured. There are no major errors to report. It is suggested to continue the research by highlighting how the stereochemistry of the double bond can influence the reactivity of stilbene at the cellular level. The stereochemistry of the entire system could be different if evaluated not only from the point of view of oxidative stress but also as an interaction in enzymatic systems.
Answer: We acknowledge and concur with your remark regarding the crucial significance of stereochemistry in profoundly modifying the reactivity and biological activity of stilbenes (combretastatin and derivatives, anticancer agents, for instance). Recent research has emphasized the importance of the Z configuration in stilbenes and hybrid derivatives, demonstrating their effectiveness in cytotoxic effects against cancer cells. Nevertheless, it is important to specify that this study solely concentrates on E-stilbenes. The selection of this strategy is based on the comprehensive investigation carried out on these compounds, their robustness, and their comparatively reduced cytotoxicity in comparison to Z-stilbenes. To appropriately represent the extent of our research, we have revised the title to " E-Stilbenes: General chemical and biological aspects, potential pharmacological activity based on the Nrf2-Pathway." We are confident that this modification will effectively communicate the precise goals and results of our research.
Some references to Z-stilbenes are not cited in the text.
Int. J. Mol. Sci. 2019, 20, 1300; doi:10.3390/ijms20061300
Pettit, G.R.; Cragg, G.M.; Herald, D.L.; Schmidt, J.M.; Lohavanijaya, P. Isolation and structure of combretastatin. Can. J. Chem. 1982, 60, 1374–1376.
Tron, G.C.; Pirali, T.; Sorba, G.; Pagliai, F.; Busacca, S.; Genazzani, A.A. Medicinal chemistry of combretastatin A4: Present and future directions. J. Med. Chem. 2006, 49, 3033–3044
Reviewer 3 Report
Comments and Suggestions for Authors
Stilbene is a class of organic compounds including molecules like resveratrol, piceatannol, pterostilbene, polydatin, and so forth, which is found in various plants and has drawn attention for its potential health benefits. They are particularly of interest in the pharmaceutical industry due to several important properties and potential pharmacological applications. This review article by Mendoca et al is to provide some basic knowledge of stilbenes (chemical structure, biosynthesis, metabolism) and assess the role of stilbenes in human metabolism and the Nrf2 activation pathway. This review is thorough, and the topic is certainly useful in the field of drug discovery. However, I have several comments which preclude this manuscript to be published in the current form.
1. The title of the review article is “Nrf2 modulation by stilbenes: possible pharmacological applications”. From what I understand, the authors intended to focus on discussing the role of stilbenes on Nrf2 pathway based on the title. In that sense, I think the authors can consider shortening and simplifying section 2 and section 3. Especially for section 3 “Stilbenes: diseases-based biological and pharmacological activities”, it is clear that stilbenes have many roles in different diseases, but some of them are irrelevant to the topic of the review.
2. Section 6 “Conclusions and future perspectives” are too thin. The authors should expand more details to discuss what other efforts should be made in the field for studying stilbenes in Nrf2 pathway. The authors should lay out all other studies needed that includes but not limited to (1) mechanistic studies including structural biology and biochemistry to elucidate how stilbenes can interact with molecules in the Nrf2 pathways. (2) Well controlled dose-response studies (3) study cell-specific and tissue-specific effect (4) Interactions with other pathways that are influenced by stilbenes (5) Long-term safety studies. Overall, a multidisciplinary approach combining biochemistry, pharmacology, cell biology, animal studies and clinical studies will be essential to comprehensively elucidate the role of stilbenes in Nrf2 pathway and their potential therapeutic applications.
3. Some tables are too long and occupies too much space, can the author provide a separate supplementary table for all of them so as to simply the tables in the main manuscript?
4. Some of the abbreviations in the figures are not explained in the figure captions. For example, in Figure 2, “PEP”, “HPP” … and others are not spelled out and explained. The authors should go over the entire manuscript and make sure all the abbreviations have been spelled out for the first time. Same thing for Figure 4.
5. In Figure 1, the authors should highlight the compounds discussed in section 5: RESV, PIC, PIN, DHS … and write the abbreviations next to the compound name as well. For example, “Resveratrol (RESV)”. Also consider deleting some compound that are not discussed in the manuscript, for simplicity.
Comments on the Quality of English LanguageThe language of this manuscript needs some editing. The manuscripts to be published on pharmaceuticals should have high language requirement.
Author Response
REVIEWER #3:
- The title of the review article is “Nrf2 modulation by stilbenes: possible pharmacological applications”. From what I understand, the authors intended to focus on discussing the role of stilbenes on Nrf2 pathway based on the title. In that sense, I think the authors can consider shortening and simplifying section 2 and section 3. Especially for section 3 “Stilbenes: diseases-based biological and pharmacological activities”, it is clear that stilbenes have many roles in different diseases, but some of them are irrelevant to the topic of the review.
Answer: We acknowledge your understanding of our objectives and concur that the title did not effectively convey our goals. The title was modified. Thanks. It is now:" E-Stilbenes: General chemical and biological aspects, potential pharmacological activity based on the Nrf2-Pathway. Our primary focus is investigating the correlation between the activation of the Nrf2 pathway and E-stilbenes. However, the chemical and metabolic characteristics are crucial for comprehending the basics of the process. Consequently, in topic 2, we delve into fundamental features regarding these biologically potent substances. Subsequently, in topic 3, we illustrate the various pharmacological activities that they may exhibit. Several studies indicate that these pharmacological effects can be attributed to the stimulation of the Nrf2 pathway, among others. In item 3, multiple pathways are discussed in addition to the Nrf2 pathway. In subject 4, we delve into a detailed examination of the Nrf2 pathway and explore how stilbenes can potentially regulate its activity. Our discussion concludes with subject 5, which provides evidence, from both clinical and pre-clinical trials, that these pharmacological activities occur only through the NRF2 pathway.
Thanks again for the crucial comment.
- Section 6 “Conclusions and future perspectives”are too thin. The authors should expand more details to discuss what other efforts should be made in the field for studying stilbenes in Nrf2 pathway. The authors should lay out all other studies needed that includes but not limited to (1) mechanistic studies including structural biology and biochemistry to elucidate how stilbenes can interact with molecules in the Nrf2 pathways. (2) Well controlled dose-response studies (3) study cell-specific and tissue-specific effect (4) Interactions with other pathways that are influenced by stilbenes (5) Long-term safety studies. Overall, a multidisciplinary approach combining biochemistry, pharmacology, cell biology, animal studies and clinical studies will be essential to comprehensively elucidate the role of stilbenes in Nrf2 pathway and their potential therapeutic applications.
Answer: We concur and express gratitude for your useful remarks. Indeed, the incorporation of these aspects in our final analysis and future outlooks bolstered the overall outcome of our findings. Topic 6 has been enhanced and improved based on those comments.
Based on the present review, it is evident that stilbenes could activate Nrf2 either directly or indirectly. They achieve this by affecting NF-κB and utilizing distinct biological pathways. This activation has implications for treating diseases or health conditions that involve inflammation and redox imbalance. Nevertheless, most documented experimental experiments were carried out using cellular or animal models. Although the positive outcomes are evident, it is crucial to conduct multiple studies that adopt a multidisciplinary approach to thoroughly investigate the role of stilbenes in the Nrf2 pathway and their potential therapeutic uses. This entails gaining a comprehensive understanding of the biology and biochemistry of how these compounds interact with other molecules, developing other laboratorial methods (enzyme-interactions investigation), conducting dose-response studies, and performing pre-clinical and clinical controlled trials, also searching for and guaranteeing the quality, stability, and purity of the chemical compounds or extracts used. These investigations are of utmost importance in exploring the therapeutic properties of natural products and their behavior in pathophysiological conditions. Consequently, investigations conducted on cellular and animal models play a vital role in assessing the safety and effectiveness of herbal medicines. These studies pave the way for further research involving human subjects and ultimately lead to the development of new formulations that can enhance the well-being and quality of life for individuals afflicted with health conditions that pose a threat to their overall health. In addition, it is crucial to consider the potential interaction of stilbenes with other metabolic or physiological processes. This contact could result in broader impacts across the body when these chemicals are used. Therefore, it is imperative to further explore this aspect. Furthermore, it is important to emphasize the essential significance of stereochemistry in the profound modification of the reactivity and biological activity of stilbenes. Therefore, this research specifically examines the arrangement of E-stilbene isomers and their hybrid derivatives, since they possess greater stability, lower cytotoxicity, and more significant biological activity as compared to the Z-stilbenes configuration. There is a deficiency in scientific and technological expertise. To ensure optimal activation of Nrf2 and/or negative regulation of NF-κB, it is crucial to establish precise dosages, delivery methods, and timing for each type of stilbene and for each specific situation. This will enable the standardization of supplementation with each chemical. Furthermore, it is imperative to assess the safety of stilbenes in the context of different situations where their potential benefits have been proposed, with the goal of offering more reliable and secure data.
- Some tables are too long and occupies too much space, can the author provide a separate supplementary table for all of them so as to simply the tables in the main manuscript?
Answer: We comprehend your apprehension. However, for this task, we thoroughly searched the existing literature to find clinical and preclinical research that explicitly examined E-stilbenes and their corresponding pharmacological effects via activating the NRF2 pathway. Tables 1 - 10 contain numerous articles that we have recognized. To enhance the comprehension and study of the readers, we have streamlined the tables by condensing them into concise and objective texts while preserving the information. Concise and pertinent details regarding each piece of evidence were listed. In addition, Pharmaceuticals does not impose any restrictions on the quantity of characters or pages, therefore enabling us to incorporate tables within the content. I trust that you comprehend our strategy to enhance the information uptake.
We acknowledge your observation and concur with this modification. The figure captions have undergone a thorough examination and necessary adjustments have been made. In addition, it is important to note that we have engaged the services of an English language proofreading agency that was recommended by MDPI. The article proceeds with the required amendments.
- Some of the abbreviations in the figures are not explained in the figure captions. For example, in Figure 2, “PEP”, “HPP” … and others are not spelled out and explained. The authors should go over the entire manuscript and make sure all the abbreviations have been spelled out for the first time. Same thing for Figure 4.
Answer: We acknowledge your observation and concur with these modifications. The figure captions have undergone a thorough examination and necessary adjustments have been made. In addition, it is important to note that we have engaged the services of an English language proofreading agency that was recommended by MDPI.
- In Figure 1, the authors should highlight the compounds discussed in section 5: RESV, PIC, PIN, DHS … and write the abbreviations next to the compound name as well. For example, “Resveratrol (RESV)”. Also consider deleting some compound that are not discussed in the manuscript, for simplicity.
Answer: We have identified and prioritized these primary stilbenes that we are now focusing our efforts on. However, it is important to note that Figure 1 exclusively includes the stilbenes that are specifically listed in the text. Topic 3 encompasses the remaining stilbenes, therefore their structural representation in picture 1. The revised figure is included in the manuscript.
Round 2
Reviewer 1 Report
Comments and Suggestions for Authors
The authors Elaine L. S. S. Mendonça et al satisfactorily answered to all my questions/suggestions. Thus I recommend its publication without farther revision.
Reviewer 3 Report
Comments and Suggestions for Authors
The author addressed all my comments. Some of the figures look a bit blur to me and please consider increase the resolution of the figures. I think the paper can be accepted after typo checking.
Comments on the Quality of English LanguageNo further comments on the language.